# A single vertebrate DNA virus protein disarms invertebrate immunity to RNA virus infection

Don B Gammon[1], Sophie Duraffour[2], Daniel K Rozelle[3], Heidi Hehnly[4], Rita Sharma[1,5], Michael E Sparks[6†], Cara C West[7], Ying Chen[1], James J Moresco[8], Graciela Andrei[2], John H Connor[3], Darryl Conte Jr.[1], Dawn E Gundersen-Rindal[6], William L Marshall[7‡], John R Yates[8], Neal Silverman[7], Craig C Mello[1,5]*

[1]RNA Therapeutics Institute, University of Massachusetts Medical School, Worcester, United States; [2]Rega Institute for Medical Research, KU Leuven, Leuven, Belgium; [3]Department of Microbiology, Boston University, Boston, United States; [4]Program in Molecular Medicine, University of Massachusetts Medical School, Worcester, United States; [5]Howard Hughes Medical Institute, University of Massachusetts Medical School, Worcester, United States; [6]Agricultural Research Service, United States Department of Agriculture, Beltsville, United States; [7]Department of Medicine, University of Massachusetts Medical School, Worcester, United States; [8]Department of Chemical Physiology, The Scripps Research Institute, La Jolla, United States

*For correspondence: craig. mello@umassmed.edu

Present address: †Multidrug-resistant Organism Repository and Surveillance Network, Walter Reed Army Institute of Research, Silver Spring, United States; ‡Merck Research Laboratories, Boston, United States

Competing interests: The authors declare that no competing interests exist.

**Abstract** Virus-host interactions drive a remarkable diversity of immune responses and countermeasures. We found that two RNA viruses with broad host ranges, vesicular stomatitis virus (VSV) and Sindbis virus (SINV), are completely restricted in their replication after entry into Lepidopteran cells. This restriction is overcome when cells are co-infected with vaccinia virus (VACV), a vertebrate DNA virus. Using RNAi screening, we show that Lepidopteran RNAi, Nuclear Factor-κB, and ubiquitin-proteasome pathways restrict RNA virus infection. Surprisingly, a highly conserved, uncharacterized VACV protein, A51R, can partially overcome this virus restriction. We show that A51R is also critical for VACV replication in vertebrate cells and for pathogenesis in mice. Interestingly, A51R colocalizes with, and stabilizes, host microtubules and also associates with ubiquitin. We show that A51R promotes viral protein stability, possibly by preventing ubiquitin-dependent targeting of viral proteins for destruction. Importantly, our studies reveal exciting new opportunities to study virus-host interactions in experimentally-tractable Lepidopteran systems.

## Introduction

Viruses represent a constantly evolving challenge to the fitness and survival of their cellular hosts. Thus, not surprisingly, investigations into virus-host interactions have produced important and fundamental new insights into both cellular and pathophysiology (*Panda and Cherry, 2012*). Invertebrate model organisms have proven useful in elucidating a wide range of host responses to infection and because many of these responses are well conserved, studies in model organisms are often directly relevant to human health (*Moser et al., 2010*; *Panda and Cherry, 2012*; *Moy et al., 2014*). Notably, studies of invertebrate antiviral RNA interference (RNAi) pathways (*Fire et al., 1998*; *Zhou and Rana, 2013*) have produced powerful tools for probing and manipulating gene function, with potential utility for direct therapeutic intervention (*Blake et al., 2012*).

Relatively small genomes, well-defined genetics, and efficient RNAi pathways make insect models attractive systems in which to study virus-host interplay (*Moser et al., 2010*; *Cherry, 2011*). Dipteran

**eLife digest** Viruses can infect species as diverse as bacteria, plants and animals, and once they have infected an organism they hijack its cells to rapidly replicate their own genetic material, which is made of DNA or RNA. Many animals, including insects, have been used as model organisms to investigate viral infections. These studies have, for example, provided insights into how viruses replicate and how they suppress their host's immune system.

One insect species that has been used in many virus-host studies is the gypsy moth. This species of moth was accidently introduced into North America from Europe in the late 1800s, and its caterpillars have become a major pest because they destroy hardwood trees and forests. Gypsy moth outbreaks are still a serious problem, but their numbers can be kept in check by using biological control strategies, such as DNA viruses. However, the response of gypsy moths to infection by RNA viruses has not been studied extensively.

Gammon et al. now show that, after being infected with one of two different RNA viruses, gypsy moth cells can slow down and eventually halt the replication of the RNA viruses. However, if the gypsy moth cells are also infected with a DNA virus, they lose their ability to restrict the replication of the RNA virus. Gammon et al. discovered that the moth's immunity to RNA virus infection is disarmed by a protein called A51R from the DNA virus. This protein increases the stability of the proteins in the RNA virus, most likely by stopping the moth from breaking them down.

The results of Gammon et al. suggest that it might be possible to use a combination of RNA viruses and the A51R protein to keep the number of gypsy moths in check.

organisms, such as *Drosophila melanogaster*, have been the primary focus of virus-host studies in invertebrates and *Drosophila* RNAi screens have greatly enhanced our understanding of how eukaryotic host factors can promote or inhibit virus replication (*Cherry, 2011*). These studies have almost exclusively focused on RNA viruses (*Xu and Cherry, 2014*). In contrast to Dipterans, most virus-host studies in the order *Lepidoptera* (moths and butterflies) have focused on DNA viruses, particularly baculoviruses (*Ikeda et al., 2013*). These studies have provided key insights into highly conserved mechanisms by which Lepidopterans combat DNA virus infection (*Ikeda et al., 2013*). Thus, while Lepidopterans provide a relevant model for studying DNA virus-host interaction they have not previously been used to probe restrictions to RNA virus replication.

The gypsy moth (*Lymantria dispar*) has been one of the most prolific North American hardwood forest pests since its accidental release in the late 1800's (*Sparks et al., 2013*). Exploring the susceptibility and responses of *L. dispar* and other Lepidopterans to virus infection is of particular importance in designing new and effective virus-based biocontrol strategies to minimize the devastating economic impact these species continue to have on the forest industry (*Sparks et al., 2013*). *L. dispar*-derived cell lines are susceptible to a wide variety of invertebrate DNA viruses, and as such, they are often used in virus-host studies (*Sparks and Gundersen-Rindal, 2011*). Interestingly, *L. dispar*-derived LD652 cells can also support a limited infection by vaccinia virus (VACV), a vertebrate poxvirus encoding a large dsDNA genome (*Li et al., 1998*). During infection of LD652 cells, VACV undergoes early gene expression, DNA replication and late gene expression, but the infection is abortive due to a defect in one or more steps of virion morphogenesis (*Li et al., 1998*). VACV entry and early gene expression have also been documented in *Drosophila* cells, however viral DNA replication and subsequent late gene expression were not detected, indicating that VACV replication is blocked earlier in its life cycle in *Drosophila* cells than in LD652 cells (*Moser et al., 2010*). Despite these limitations, RNAi screening of VACV-infected *Drosophila* cells has identified multiple host factors required for VACV entry in eukaryotic hosts (*Moser et al., 2010*). Thus, the LD652 cell culture system provides a unique model in which to explore multiple aspects of vertebrate DNA virus biology, including basic replication strategies and suppression of host immune pathways by viral proteins.

The extent of RNA virus studies in *Lepidoptera* is limited compared to DNA virus studies and largely restricted to non-enveloped dsRNA and (+)-sense ssRNA viruses such as cypoviruses (*Hill et al., 1999*), iflaviruses (*van Oers, 2010*) and tetraviruses (*Short and Dorrington, 2012*). These viruses only infect invertebrate hosts and several cannot productively replicate in cultured cells (*Short and Dorrington, 2012*). Furthermore, to our knowledge, (−)-sense ssRNA viruses have not been previously reported to

productively infect Lepidopteran hosts. A new model system for studying RNA viruses in Lepidopteran hosts may be useful in the design of new biocontrol agents for pest species and improve our understanding of RNA virus-induced disease in vertebrates.

Here we explore RNA virus-Lepidopteran host interactions by infecting LD652 cells with the (−)-sense ssRNA vesicular stomatitis virus (VSV) or the (+)-sense ssRNA Sindbis virus (SINV), both of which replicate in a wide range of invertebrate and vertebrate hosts (*Letchworth et al., 1999*; *Xiong et al., 1989*). We unexpectedly found that LD652 cells restrict both VSV and SINV replication after virus entry. Using RNAi to knock down the expression of candidate *L. dispar* antiviral immunity factors, we show that specific RNAi and innate immune pathway components restrict RNA virus replication. We also uncover a role for the Lepidopteran ubiquitin-proteasome system (UPS) in restricting RNA virus replication. Surprisingly, co-infection with VACV strongly suppressed this restriction, suggesting that VACV encodes one or more factors that promote RNA virus replication. Using RNAi and genetic techniques, we found that the highly conserved, and previously uncharacterized, VACV *A51R* gene product is sufficient to alleviate the LD652 cell restriction to VSV and SINV replication. Interestingly, A51R formed aggregate- and filament-like structures that colocalize with microtubules (MTs) and protected MTs from depolymerization. Using alanine mutagenesis, we further show that an A51R point mutant with reduced RNA virus rescue ability still forms filamentous structures and stabilizes MTs, suggesting that A51R functions, in addition to MT stabilization, are required for disarming Lepidopteran antiviral immunity. Using mass spectrometry-based techniques, we found that A51R co-immunoprecipitates with several host proteins, including ubiquitin (Ub). Using radiolabeling and immunoblotting, we show that A51R does not affect viral mRNA translation rates but does promote virus protein stability, possibly by inhibiting Ub-dependent host targeting of viral proteins for degradation. Importantly, we show that A51R is also required for efficient replication of VACV in vertebrate cells and for pathogenesis in mice, indicating that A51R is a VACV virulence factor. Collectively, our findings demonstrate the utility of Lepidopteran systems for the study of RNA- and DNA virus host interactions and shed light on how this economically-important order of insects restricts virus replication.

## Results

### RNA virus restriction in *L. dispar* cells is relieved by VACV co-infection

To determine the susceptibility of *L. dispar* cells to RNA virus infection, we challenged LD652 cells with recombinant strains of VSV and SINV that express either green fluorescent protein (GFP) or luciferase (LUC) from viral promoters. In single infections with VSV-GFP (*Kato et al., 2005*) or SINV-GFP (*Cristea et al., 2006*), we found that, even at a high multiplicity of infection (MOI) of 10, <4% of LD652 cells exhibited GFP fluorescence by 96 hr post-infection (hpi) (*Figure 1A,B*). A previous report indicated that VACV enters LD652 cells and reaches the stage of late gene expression but ultimately fails to complete virion morphogenesis (*Li et al., 1998*). VACV, like other poxviruses, encodes numerous immunomodulatory proteins that inhibit a wide variety of host antiviral pathways (*Smith et al., 2013*). We therefore wondered if VACV co-infection (although abortive) might nevertheless overcome an immune response in LD652 cells that restricts RNA virus replication. Consistent with this idea, when LD652 cells were co-infected with VSV-GFP or SINV-GFP and the Western Reserve (WR) strain of VACV (VACV-WR), the number of GFP-positive cells increased to ~77% (VSV-GFP) and ~45% (SINV-GFP) by 96 hpi (*Figure 1A,B*).

We next employed VSV-LUC (*Cureton et al., 2009*) and SINV-LUC (*Cook and Griffin, 2003*) recombinant strains along with sensitive chemiluminescence-based LUC assays to detect and measure virus gene expression. Using this strategy, we detected small, ~3–5-fold increases in light units (LU) during VSV-LUC (*Figure 1C*) or SINV-LUC (*Figure 1D*) infection of LD652 cells between 8 and 24 hpi, with levels plateauing by 24 hpi. Notably, these single infections still produced LU levels ~10-fold above mock-infected cells by 24 hpi. Co-infection of VSV-LUC or SINV-LUC with VACV-WR initially yielded LU levels similar to those seen in single infection by 8 hpi; however, LU readings increased logarithmically by 24–48 hpi (*Figure 1C,D*).

The trends observed in the LUC assays were further confirmed by immunoblotting for LUC and other viral proteins (*Figure 1E,F*). Although we detected LUC enzymatic activity in VSV-LUC-infected cells in the absence of VACV-WR co-infection, we were typically unable to detect LUC protein on immunoblots under these conditions. We did, however detect a small amount of VSV Matrix (M) structural protein (*Figure 1E*). Enhancement of LUC and SINV E1 capsid protein expression during VACV-WR co-infection with SINV-LUC was also confirmed by immunoblot (*Figure 1F*). Importantly, measurement

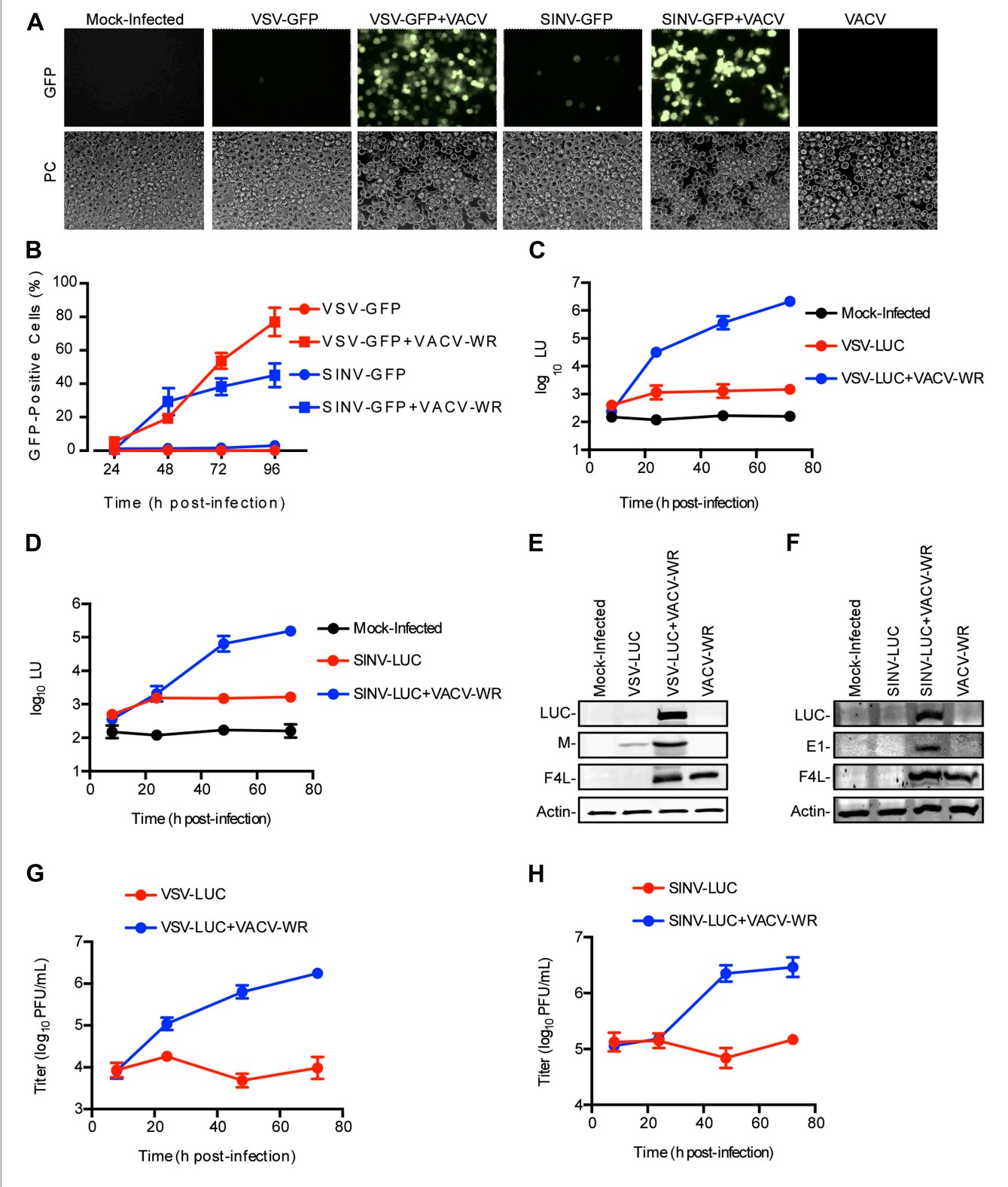

**Figure 1**. Restriction of RNA virus replication in LD652 cells is relieved by VACV co-infection. (**A**) GFP fluorescence (top) and phase contrast images (PC, bottom) of infected LD652 cells at 96 hpi. Images are shown at 20X magnification. (**B**) Percentage of GFP-positive LD652 cells from experiments in (**A**). (**C** and **D**) LUC assay [arbitrary light units (LU)] of lysates from mock-infected cells or cells infected with VSV-LUC (**C**) or SINV-LUC (**D**), in the absence or
*Figure 1. Continued on next page*

*Figure 1. Continued*

presence of VACV-WR. Mock-infected data are identical in (**C**) and (**D**). (**E** and **F**) Immunoblot of LUC, VSV M (**E**) or SINV E1 (**F**), VACV F4L, and cellular actin proteins in lysates from (**C**) and (**D**) 72 hpi. (**G** and **H**) VSV-LUC (**G**) or SINV-LUC (**H**) titers, (plaque-forming units (PFU)/ml) in culture supernatants from (**C**) and (**D**), respectively. Quantitative data represent means (±SEM). See also *Figure 1—figure supplement 1*.

The following figure supplements are available for figure 1:

**Figure supplement 1**. Restriction of RNA virus replication in *L. dispar* cells occurs at a step post-entry.

of VSV-LUC and SINV-LUC titers from LD652 cultures also reflected LUC activity, remaining unchanged over time in single infections but increasing during VACV-WR co-infection (*Figure 1G,H*). Collectively, these data show that LUC activity mirrors RNA virus gene expression and virion production thus providing a sensitive measure of RNA virus replication in LD652 cells.

## Multiple host factors restrict RNA virus replication after entry into LD652 cells

The low level, but above background, LUC activity detected in VSV-LUC and SINV-LUC singly-infected LD652 cultures suggested that these viruses are blocked at one or more steps post-entry. Consistent with this, we detected VSV and SINV capsids inside singly-infected LD652 cells using confocal microscopy (*Figure 1—figure supplement 1A,B*). Furthermore, at 8 hpi RT-PCR analysis of VSV- and SINV-infected cells detected a similar level of viral transcripts with or without VACV-WR co-infection (*Figure 1—figure supplement 1C,D*). Remarkably, we could also rescue VSV and SINV gene expression (albeit to reduced levels) by the addition of VACV-WR as late as 24 hpi with RNA virus (*Figure 1—figure supplement 1E,F*). Together these data indicate that RNA virus restriction occurs post-entry.

To ask if host transcription was required to resist RNA virus infection, we treated VSV-LUC and SINV-LUC-infected LD652 cells with increasing doses of actinomycin D (ActD) (*Black and Brown, 1968*) and then measured LUC activity 48 hpi. We chose an ActD dose range such that the highest dose (0.1 µg/ml) reduced cell viability by ~50% after 48 hr of treatment (*Figure 2A*). We found that the higher ActD doses enhanced viral gene expression by as much as ~100-fold compared to control treatments (*Figure 2B*). These results suggest that host cell transcription is required for LD652 resistance to RNA virus infection and that LD652 cells express antiviral immunity factors. It should be noted that ActD treatment can induce apoptosis in invertebrate cells (*Wang et al., 2008*) and previous studies have found apoptosis induction to enhance SINV replication in mosquitoes (*Wang et al., 2012*), thus it is possible that the enhanced viral replication observed in the presence of ActD may be in part due to apoptosis induction in LD652 cells.

To identify viral-resistance pathways in *L. dispar* cells, we used our published (*Sparks and Gundersen-Rindal, 2011*), as well as unpublished mRNA sequencing (mRNA-seq) data to identify candidate *L. dispar* homologs of eukaryotic innate immunity factors. We then prepared dsRNA to silence candidate immune-related factors by RNAi in LD652 cells. After RNAi knockdown, we challenged cells with VSV-LUC and SINV-LUC infection. As a positive control for knockdown, we used dsRNAs targeting *luc* sequences. Knockdowns of four transcripts, encoding homologs of the RNAi pathway proteins AGO2 and Dicer-2 (*Kingsolver et al., 2013*), the Nuclear Factor kappa B (NF-κB)-homolog Relish (*Dushay et al., 1996*), and the E2 ubiquitin (Ub)-conjugating enzyme Effete (*Treier et al., 1992*), increased LUC expression ~10-fold in both VSV-LUC- and SINV-LUC-infected cells. RNAi-targeting three Ub-encoding genes, including Polyubiquitin, Ub-RPL40 and Ub-RPS27A increased LUC expression nearly 1000-fold in VSV-LUC-infected cells (*Figure 2C*) and 10-fold in SINV-LUC-infected cells (*Figure 2D*). Ub-RPL40 and Ub-RPS27A encode highly conserved fusion proteins that are cleaved by endogenous proteases into free Ub and either ribosomal protein L40 (RPL40) or ribosomal protein S27A (RPS27A), components of the 60S and 40S ribosomal subunits, respectively (*Finley et al., 1989*).

The dramatic effect of Ub RNAi on LUC expression in VSV-LUC-infected cells was not caused by increased LUC protein stability due to inhibition of the Ub-proteasome system (UPS) (See *Figure 2—figure supplement 1*). We found that viral-driven, but not plasmid-driven (p166-LUC), LUC expression was enhanced in experiments aimed at inhibiting the UPS, including: (i) exposure to Ub RNAi (*Figure 2—figure supplement 1A*), (ii) treatment with inhibitors of either E1 Ub-activating enzymes (PYR-41; *Figure 2—figure supplement 1B*) (*Yang et al., 2007*), or the proteasome (MG132; *Figure 2—figure supplement 1C*) (*Lee and Goldberg, 1998*), and finally, by (iii) transient expression of a wild-type (but not a catalytically inactive) coronavirus-encoded deubiquitinase (PLpro; *Figure 2—figure supplement 1D*)

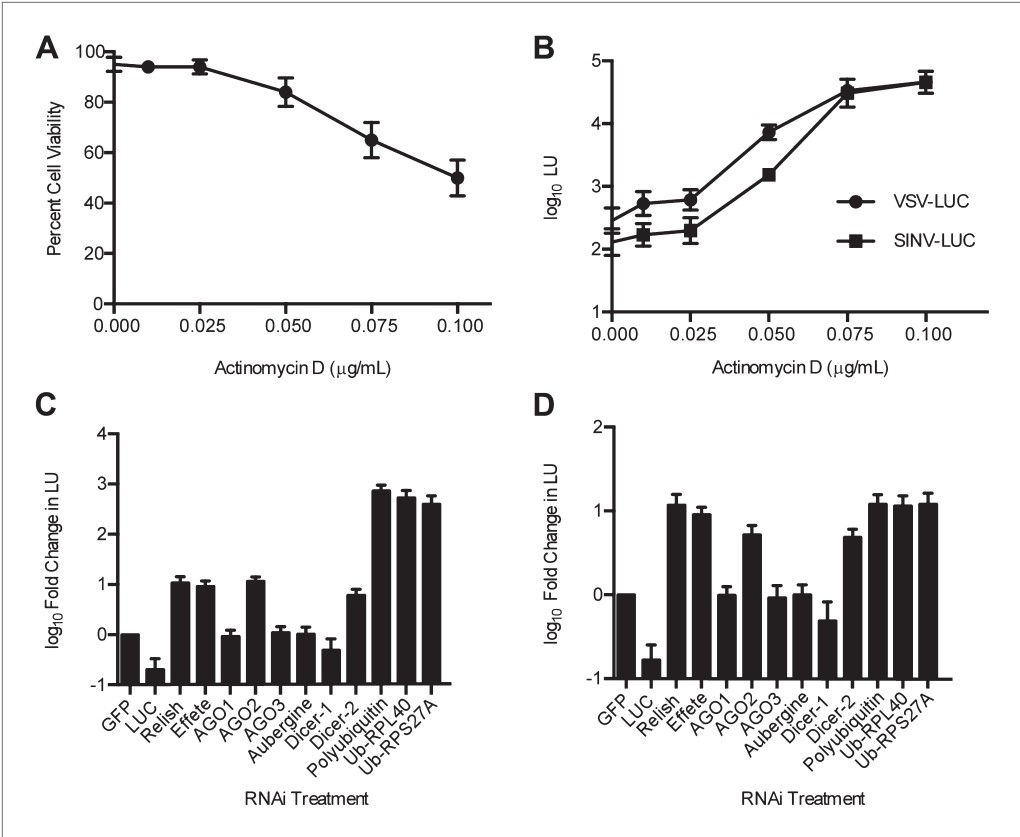

**Figure 2**. Host transcription and RNAi-, NF-κB-, and ubiquitin-related factors restrict RNA virus replication in LD652 cells. (**A**) Trypan blue exclusion assay to measure cell viability in the presence of ActD (48 hr). (**B**) Effect of ActD on virus LUC activity (in LU) 48 hpi. (**C** and **D**) LUC activity (LU) in lysates from cells 48 hpi with either VSV-LUC (**C**) or SINV-LUC (**D**) and after RNAi of *L. dispar* transcripts relative to LU generated in GFP (control) RNAi treatments. Data represent means (+SEM). See also *Figure 2—figure supplement 1*.

The following figure supplements are available for figure 2:

**Figure supplement 1**. RNA virus gene expression is enhanced upon inhibition of the ubiquitin-proteasome system in *L. dispar* cells.

---

(***Barretto et al., 2005***). Taken together, these findings suggest that host restriction of RNA virus replication in LD652 cells is mediated by multiple host factors including those involved in RNAi-, NF-κB, and Ub-related pathways.

## VACV A51R promotes rescue of RNA virus infection in LD652 cells

We next sought to identify the VACV gene(s) responsible for relieving the LD652 restriction to RNA virus infection using several complementary lines of investigation. To test if VACV binding and entry into LD652 cells is sufficient to rescue RNA virus infection in the absence of VACV gene expression, we co-infected LD652 cells with VSV or SINV and a heat-inactivated strain of VACV expressing the late A5L core protein fused to GFP (A5L-GFP) (***Carter et al., 2003***). Heat inactivated VACV, which can enter cells but cannot express its genes (***Dales and Kajioka, 1964***), failed to rescue VSV and SINV replication (***Figure 3A***), indicating that VACV gene expression was required for rescue. Next we treated co-infected cells with the viral DNA polymerase inhibitor cytosine arabinoside (AraC) at a dose that blocks viral DNA replication and subsequent late poxvirus gene expression (***Li et al., 1998***). We confirmed that AraC treatments blocked late VACV gene expression by immunoblotting for A5L-GFP in the absence or presence of AraC (data not shown). We found that inhibiting post-replicative VACV gene expression did not prevent rescue of either VSV or SINV (***Figure 3B***), narrowing the RNA virus rescue activity to one or more of the 118 early VACV genes (***Yang et al., 2010***).

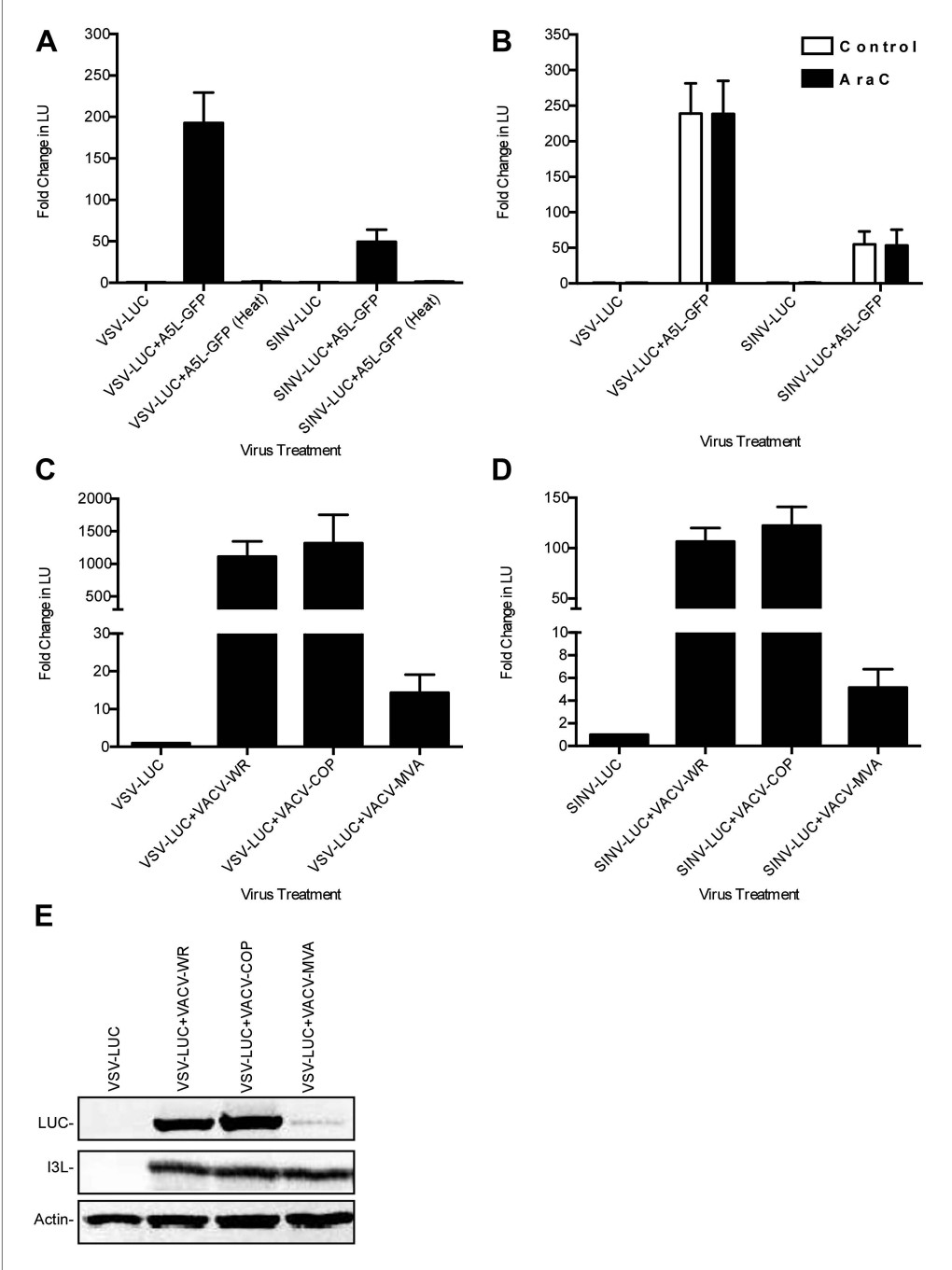

**Figure 3**. Characterization of VACV-dependent RNA virus rescue in LD652 cells. (**A**) Effect of heat inactivation of VACV strain A5L-GFP on RNA virus rescue 48 hpi by LUC assay (in LU). LU generated from co-infection lysates and are plotted as 'fold change' with respect to single infection conditions. (**B**) Effect of AraC (200 µg/ml) treatment on A5L-GFP-mediated RNA virus rescue 48 hpi. Fold change in LU was calculated as in (**A**). (**C** and **D**) Relative LUC activity in lysates from cells infected with VSV-LUC (**C**) or SINV-LUC (**D**) and co-infected with various VACV strains and in the presence of AraC for 72 hr. Fold change in LU was calculated as in (**A**) VACV strain Copenhagen (VACV-COP); modified VACV Ankara (VACV-MVA). (**E**) Immunoblot of lysates from (**C**) for LUC, I3L, and actin. Data represent means (+SEM).

We next screened a collection of VACV mutants for their ability to rescue RNA viruses in LD652 cells. We found that the attenuated modified VACV strain Ankara (VACV-MVA) (*Antoine et al., 1998*) failed to fully rescue either VSV or SINV as compared to either VACV-WR or Copenhagen strains

(*Figure 3C,D*), despite similar levels of viral gene expression by all three VACV strains (*Figure 3E*). These results suggested that VACV-MVA is defective for one or more factors required for RNA virus rescue in LD652 cells.

The above findings narrowed the number of candidates to ~30 genes that are deleted or truncated in VACV-MVA relative to VACV-WR (*Meisinger-Henschel et al., 2007*). We then used RNAi to knock down each of the candidate VACV genes during VSV and SINV co-infection with VACV-WR (*Figure 4A*). RNAi of a single, uncharacterized VACV gene, *A51R*, caused RNA virus gene expression to drop below an arbitrary cut off of 50% of the control (GFP) RNAi treatment. The A51R knockdown reduced virus gene expression by ~95% (VSV) and by ~69% (SINV) (*Figure 4B*, *Figure 4—figure supplement 1*).

To confirm the necessity of *A51R* for VACV rescue of RNA virus infection, we created a recombinant VACV-WR strain with a deletion of the *A51R* gene (ΔA51R) (*Figure 4—figure supplement 2A*). Importantly, we found that co-infection with ΔA51R resulted in a 100-fold reduction in VSV gene expression relative to co-infection with VACV-WR (*Figure 4C*). VSV was rescued by co-infecting with revertant VACV strains (*Figure 4C*), expressing a Flag-tagged *A51R* gene (*Flag-A51R*) reintroduced into either the *A51R* locus (ΔA51R^FA51R^) or the VACV thymidine kinase (*J2R*) locus (ΔJ2R/ΔA51R^FA51R^). In addition, we reverted the ΔA51R strain with a Flag-tagged form of the VACV-MVA *A51R* gene (*Flag-MVAA51R)* introduced into the *J2R* locus (ΔJ2R/ΔA51R^FMVAA51R^). Flag-A51R and Flag-MVAA51R proteins migrated as single bands of ~38 and 35 kDa, respectively, on immunoblots (*Figure 4—figure supplement 2B,C*). The lower molecular weight of the VACV-MVA protein is likely due to a C-terminal truncation of ~20 amino acids that results from a genomic deletion that occurred during VACV-MVA passage in culture (*Antoine et al., 1998*). Furthermore, consistent with the idea that the VACV-MVA A51R protein is not functional, we found that the ΔJ2R/ΔA51R^FMVAA51R^ strain failed to fully rescue VSV (*Figure 4C*). Importantly, immunoblot of lysates from a time course of A51R^FA51R^ infection confirmed that Flag-A51R is expressed early in VACV infection, as it was detected by 2 hpi in the absence or presence of AraC (*Figure 4—figure supplement 2D*). Together these results indicate that A51R is an early VACV protein necessary to relieve the restriction of VSV in LD652 cells.

A51R was also required for full rescue of SINV replication by VACV (*Figure 4D*). However, we noticed that ΔA51R and ΔJ2R/ΔA51R^FMVAA51R^ strains exhibited only a moderate fourfold reduction in their ability to promote SINV gene expression as compared to 100-fold reductions for VSV (*Figure 4C,D*). These findings suggest that other VACV factors in addition to A51R promote SINV replication in LD652 cells.

## Poxvirus A51R proteins are sufficient to promote RNA virus replication in LD652 cells

We next wished to determine if expression of A51R was sufficient to rescue VSV and SINV infection in LD652 cells in the absence of other VACV components. Therefore, we cloned *Flag-A51R* into a p166 expression vector and transiently transfected LD652 cells. Although only ~40–60% of cells expressed detectable Flag-A51R protein (data not shown), we observed an ~60-fold increase in VSV and 12-fold increase in SINV gene expression relative to control transfections with empty p166 vector or vector encoding Flag-tagged GFP (Flag-GFP) (*Figure 4E*). Immunoblots confirmed similar levels of Flag-GFP and Flag-A51R expression in these lysates (*Figure 4F*). VSV and SINV titers were similarly enhanced by Flag-A51R transfection (*Figure 4G–H*). These results indicate that Flag-A51R expression is sufficient to overcome the host restriction to RNA virus replication in the absence of other VACV proteins. The stronger A51R-dependent rescue of VSV compared to SINV led us to focus on VSV infection assays in our further studies of A51R-mediated rescue.

We next tested if VSV replication affected LD652 cell viability. Transfection of Flag-GFP or Flag-A51R constructs into LD652 cells did not lead to notable changes in cell viability over a 7 day time course. However, when transfected cells were subsequently infected with VSV-LUC, cell viability began to drop by 96 hpi, ultimately reaching cell viabilities of ~50% by day 7 of infection (*Figure 4I*).This reduced cell viability was not observed when Flag-GFP-transfected cells were infected with VSV-LUC, suggesting that active viral replication was required to induce cell death (*Figure 4I*).

*A51R* genes are found in most vertebrate poxvirus genera (*Table 1*) yet absent in the genomes of entomopoxviruses (not shown). We generated constructs to express some of these *A51R* homologs, with amino acid identities ranging from 90% to 30% relative to VACV-WR A51R (*Table 1*), and asked whether the ability of A51R to rescue VSV replication is conserved between poxvirus genera. Strikingly, we found that A51R from each poxvirus enhanced VSV gene expression to a similar level (*Figure 5*).

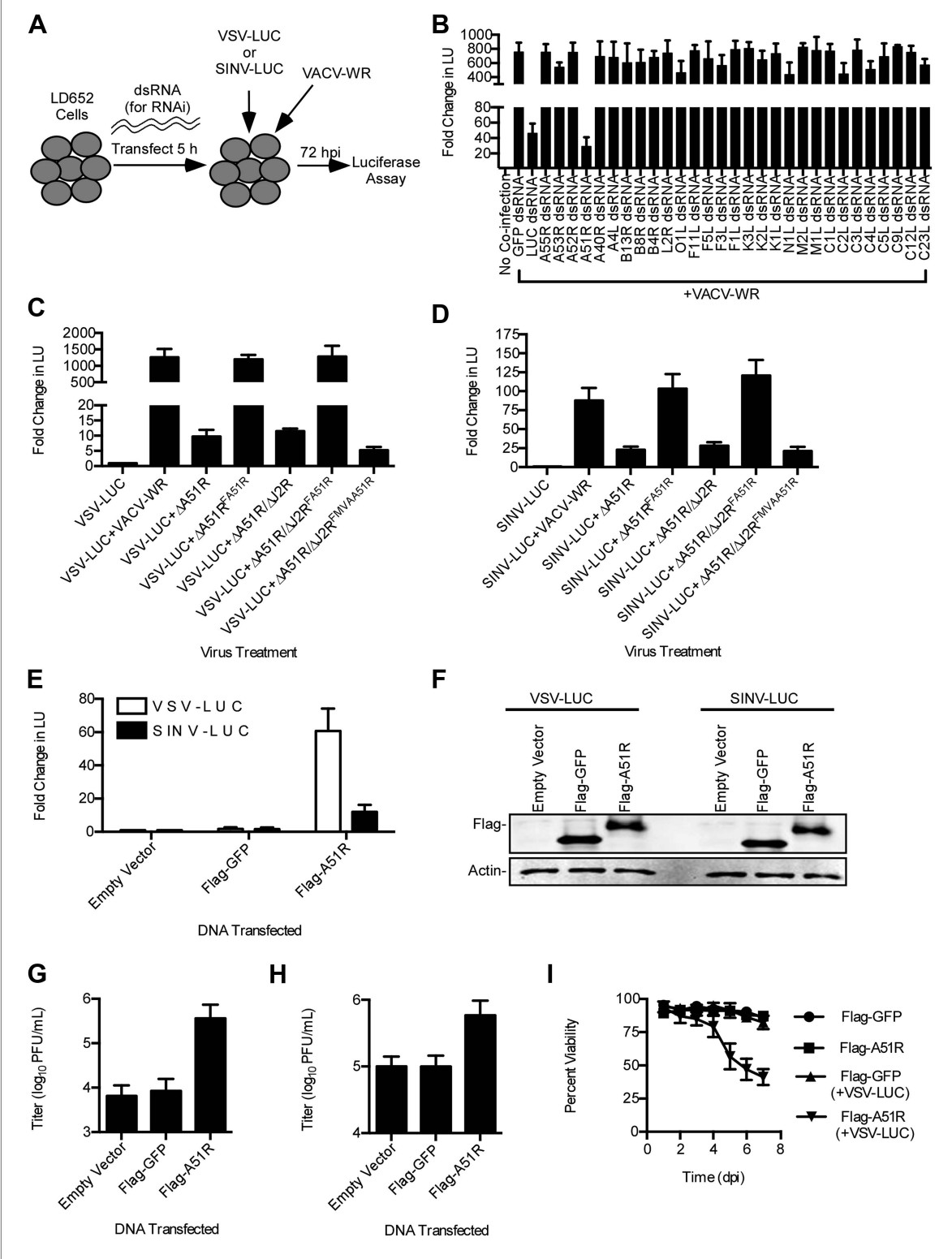

**Figure 4**. A51R is a major determinant of VACV-mediated rescue of RNA virus replication in LD652 cells. (**A**) Strategy to identify VACV 'rescue' factors using dsRNA-mediated RNAi in *L. dispar* cells. (**B**) Relative LUC activity 72 hpi in lysates from cells infected with VSV-LUC and VACV-WR after the indicated RNAi treatment. Fold change in LU was calculated as in *Figure 3A*. (**C** and **D**) Relative LUC activity in lysates from cells co-infected with recombinant VACV strains and VSV-LUC (**C**) or SINV-LUC (**D**) in the presence of AraC (200 µg/ml) for 72 hr. (**E**) Relative LUC activity in lysates of cells

*Figure 4. Continued on next page*

*Figure 4. Continued*

transfected with empty p166 vector, Flag-GFP, or Flag-A51R for 24 hr and then infected with VSV-LUC or SINV-LUC for 48 hr. Fold change in LU are relative to LU obtained from empty vector treatment. (**F**) Immunoblot of Flag-GFP (lower band) and Flag-A51R (upper band) from lysates in (**E**). Actin served as a loading control. (**G**–**H**) VSV-LUC (**G**) and SINV-LUC (**H**) titers from experiments in (**E**). (**I**) Cell viability after VSV infection as measured by trypan blue exclusion assay. Cells were transfected for 24 hr with either Flag-GFP or Flag-A51R vectors and then mock-infected or infected with VSV-LUC. Data represent means (+SEM). See also *Figure 4—figure supplements 1 and 2*.

The following figure supplements are available for figure 4:

**Figure supplement 1**. Identification of VACV A51R as a determinant of SINV rescue in *L. dispar* cells.

**Figure supplement 2**. Characterization of A51R-related recombinant VACV strains.

**Table 1.** Differential conservation of *Chordopoxirinae A51R* genes

| Genus | A51R | Example species (% amino acid identity to VACV-WR A51R) |
|---|---|---|
| *Orthopoxvirus* | + | VACV* |
| | | HSPV† (96) |
| | | CPXV (94) |
| | | ECTV (93) |
| | | VARV (92) |
| *Suipoxvirus* | + | SPXV (33) |
| *Yatapoxvirus* | + | TANV (35) |
| | | YLDV (35) |
| *Leporipoxvirus* | + | MYXV (35) |
| | | SFV (35) |
| *Capripoxvirus* | + | GTPV (32) |
| | | SPPV (30) |
| | | LSDV (29) |
| *Cervidpoxvirus* | + | DPV (31) |
| *Parapoxvirus* | + | ORFV (22) |
| *Molluscipoxvirus* | – | MCV |
| *Avipoxvirus* | – | FPV |
| | | CNPV |
| Unclassified | – | CRV |

*VACV strains MVA (**Antoine et al., 1998**) and Dryvax (**Qin et al., 2011**) contain a truncated and fragmented *A51R* gene, respectively.
†HSPV contains a fragmented *A51R* gene (**Tulman et al., 2006**).
'+' Indicates presence, and '−' indicates absence of *A51R* gene in viral genomes within each genus. Abbreviations: VACV, vaccinia virus; HSPV, horsepox virus; TATV, taterapox virus; VARV, variola virus; SPXV, swinepox virus; TANV, tanapox virus; yaba-like disease virus; MYXV, myxoma virus; SFV, Shope fibroma virus; GTPV, goatpox virus; SPPV, sheeppox virus; LSDV, lumpy skin disease virus; DPV, deerpox virus; FPV, fowlpox virus; CNPV, canarypox virus; MCV, molluscum contagiosum; ORFV, orf virus; CRV, crocodilepox virus.

Thus, the ability of A51R to overcome the LD652 restriction to VSV infection is a conserved function of distantly-related A51R homologs.

## A51R promotes RNA virus infection in multiple Lepidopteran hosts

The ability of a vertebrate poxvirus to rescue VSV replication in LD652 cells was so unexpected that we asked whether other cell lines from *L. dispar* or other Lepidopteran species similarly restrict VSV replication. Indeed, we found that A51R expression relieved restriction to VSV replication in each of the three additional Lepidopteran cell lines tested, including an *L. dispar* embryonic cell line (LdEP) (*Figure 6A*), a *Spodoptera frugiperda*-derived cell line (Sf9) (*Figure 6B,C*), and a *Manduca sexta*-derived cell line (GV-1) (*Figure 6D,E*).

In contrast, A51R expression did not enhance VSV replication in *D. melanogaster*-derived DL1 cells (*Figure 6—figure supplement 1A*), despite significant expression of Flag-A51R protein (*Figure 6—figure supplement 1B*). VSV gene expression in DL1 cells was also not enhanced by co-infection with VACV-WR (*Figure 6—figure supplement 1C,D*). Thus, if *Drosophila* cells do have restrictions targeted by A51R, these restrictions are not sufficient to measurably check VSV replication in these experiments.

## A51R proteins localize to MT networks and protect them from depolymerizing agents

Bioinformatic analyses of A51R proteins with public protein databases failed to reveal any functional domains or motifs that might provide insight into A51R function. Therefore we used confocal microscopy to begin A51R characterization by examining A51R subcellular localization. We found that Flag-A51R, expressed under its natural promoter from the ΔA51R^FA51R virus, formed both aggregate-like as well as filamentous structures in the cytoplasm of African green

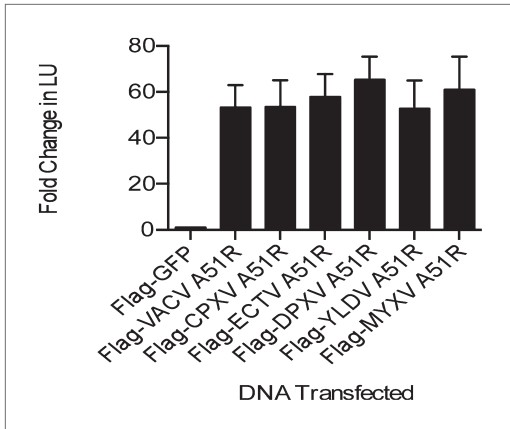

**Figure 5**. A51R proteins from disparate poxviruses rescue RNA virus replication in LD652 cells. Relative LUC activity in lysates of cells expressing the indicated A51R construct and infected with VSV-LUC for 48 hr. Fold change in LU are relative to LU obtained in Flag-GFP treatments. CPXV, cowpox virus; ECTV, ectromelia virus; DPXV, deerpox virus; YLDV, Yaba-like disease virus; and MYXV, myxoma virus. Data represent means (+SEM).

monkey-derived (BSC-40) cells. This staining pattern significantly overlapped with that of cellular tubulin, suggesting that A51R associates with a subset of MTs (*Figure 7—figure supplement 1A*). Indeed deconvolution of images from these experiments revealed Flag-A51R staining localized on intact MT tracks (*Figure 7—figure supplement 1B*).

Consistent with the idea that A51R not only co-localizes with MTs but also alters their properties, we observed enhanced co-localization on A51R-dependent, drug-resistant MT structures in infected cells treated with the MT-depolymerizing drug nocodazole. In the presence of nocodazole, MT structures were essentially absent in mock-infected and ΔA51R-infected cells. However filamentous, drug-resistant 'MT pieces' were abundant in ΔA51R^FA51R^-infected BSC-40 cells (*Figure 7—figure supplement 1C*).

Nocodazole-resistant MT pieces were previously observed in mammalian cells infected with VACV-WR but not with VACV-MVA (*Ploubidou et al., 2000*; *Schepis et al., 2006*). We therefore examined Flag-MVAA51R protein localization in nocodazole-treated cells and found that, although Flag-MVAA51R staining stained small punctate structures throughout the cytoplasm that overlapped with tubulin, large drug-resistant MT-like structures failed to form (*Figure 7—figure supplement 2A*). These results suggest that Flag-MVAA51R protein is deficient in its ability to both form filament-like structures and to protect MTs from depolymerization.

To ask if Flag-A51R localization to and stabilization of MTs requires other VACV proteins, we transfected BSC-40 cells with a plasmid expressing Flag-A51R. We found that Flag-A51R expression was sufficient for both localization to, and stabilization of, MTs (*Figure 7A*). Moreover, using similar assays, we found that A51R homologs from other poxviruses also co-localize with and protect MTs from depolymerization (*Figure 7—figure supplement 2B*). Finally, in LD652 cells, as in vertebrate cells, Flag-A51R formed aggregates and filamentous structures that co-localized with and protected MTs from depolymerization by the MT-depolymerizing agent vincristine (*Figure 7B*). Taken together, our findings suggest that the ability of A51R to interact with the host MT cytoskeleton is a conserved property of A51R proteins.

## A51R-dependent MT stabilization is insufficient for full RNA virus rescue

Based on the conservation of RNA virus rescue and MT stabilization functions of poxvirus A51R proteins, we wondered if these two functions could be separated. Therefore we performed systematic alanine mutagenesis of A51R residues that were directly conserved or conserved based on charge. After screening over 30 point mutations throughout the VACV A51R protein, we found that almost all of the mutants displayed significantly reduced protein stability, rendering virus rescue and MT-related functions difficult to assess (data not shown). However, alanine substitution of residue R321, which lies within the C-terminal region of A51R that is missing in VACV-MVA A51R (*Figure 8A*), produced a suitable mutant for analysis. Flag-A51R^R321A^ proteins displayed a reduced VSV rescue phenotype when compared to Flag-A51R proteins (*Figure 8B*) despite similar levels of expression (*Figure 8C*). Importantly, when expressed in LD652 cells in the presence of vincristine, both Flag-A51R (*Figure 8D*; *Video 1*) and Flag-A51R^R321A^ (*Figure 8D*; *Video 2*) formed filamentous structures that overlapped with tubulin staining (*Figure 8D*). These results suggest that MT stabilization alone is insufficient to promote full RNA virus rescue.

## A51R associates with Ub in Lepidopteran cells and promotes viral protein stability

The previous results suggested that A51R may provide functions, beyond MT stabilization, that promote RNA virus replication. In order to explore A51R protein complexes we used highly-sensitive

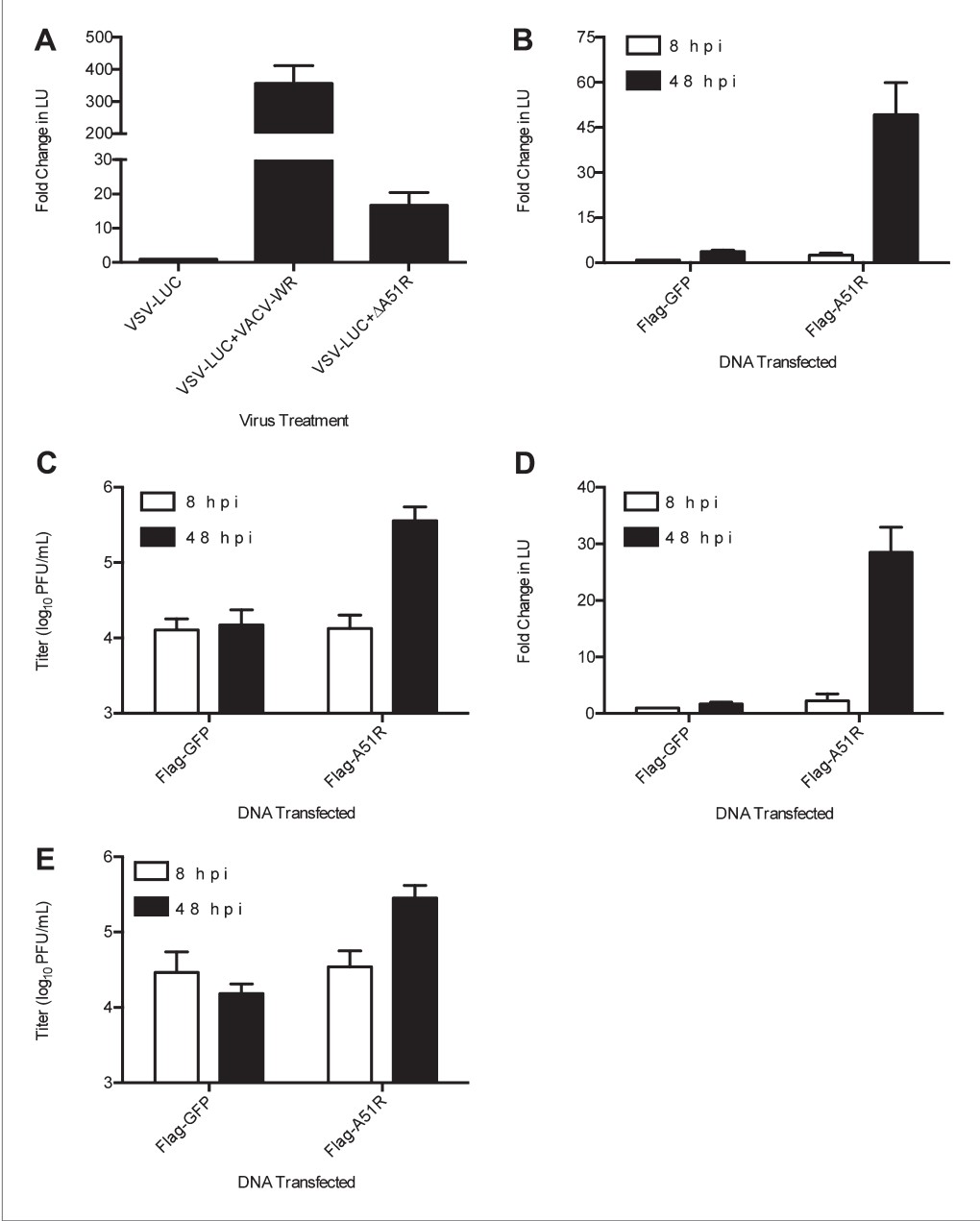

**Figure 6**. A51R relieves RNA virus restriction in multiple Lepidopteran hosts. (**A**) Relative LUC activity in lysates of *L. dispar*-derived embryonic (LdEP) cells co-infected with VSV-LUC and VACV-WR or ΔA51R strains. Fold changes in LU are relative to LU obtained from VSV-LUC (8 hpi) lysates. (**B**) Relative LUC activity in lysates of *Spodoptera frugiperda*-derived Sf9 cells expressing Flag-GFP or Flag-A51R and infected with VSV-LUC. Fold change in LU are relative to LU obtained from Flag-GFP (8 hpi) lysates. (**C**) Virus titer of supernatants from (**B**). (**D**) Relative LUC activity in lysates of *Manduca sexta*-derived GV-1 cells expressing Flag-GFP or Flag-A51R and infected with VSV-LUC. Fold change in LU were calculated as in (**B**). (**E**) Virus titer of supernatants from (**D**). Data represent means (+SEM). See also *Figure 6—figure supplement 1*.

The following figure supplements are available for figure 6:

**Figure supplement 1**. Effect of Flag-A51R expression or VACV-WR co-infection on VSV-LUC gene expression in Drosophila cells.

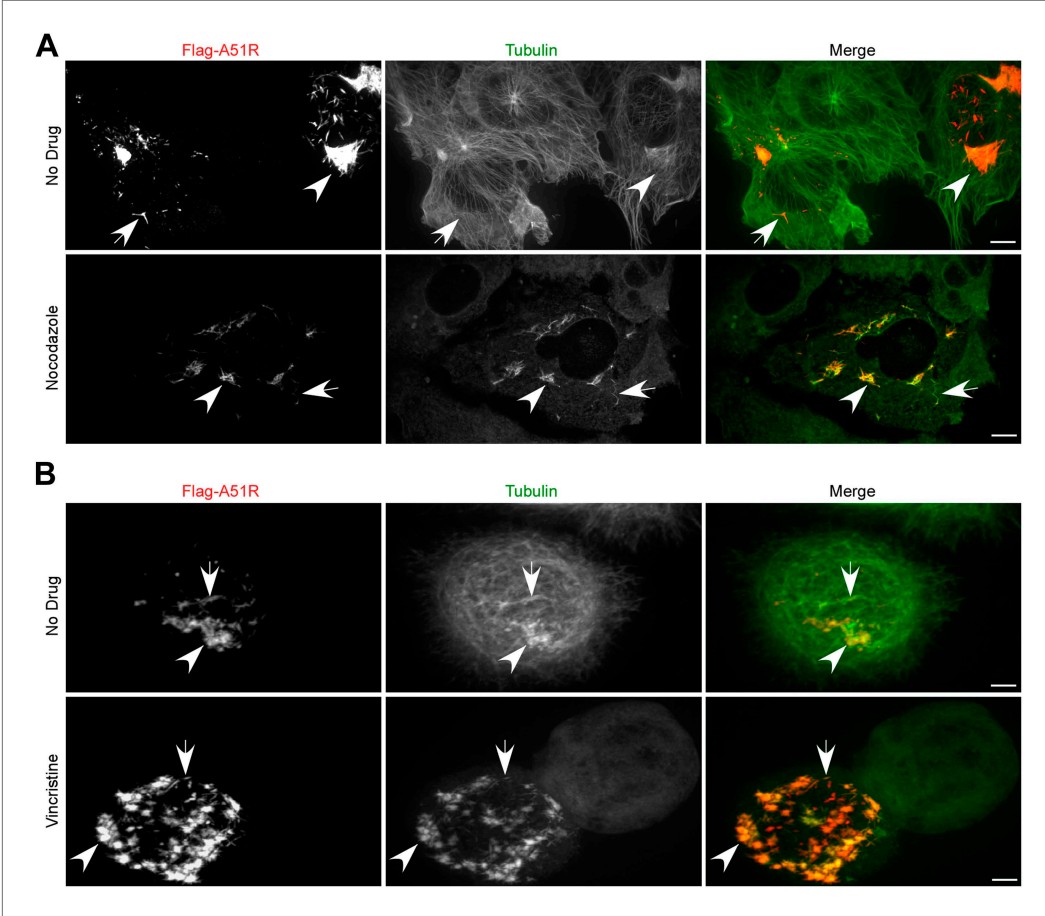

**Figure 7**. Poxvirus A51R proteins localize with and stabilize host microtubules. (**A**) Immunofluorescence of Flag-A51R (red) and tubulin (green) in BSC-40 cells in the presence or absence of nocodazole 24 hr post-transfection of Flag-A51R vector. (**B**) Immunofluorescence of Flag-A51R (red) and tubulin (green) in LD652 cells in the presence or absence of vincristine 24 hr post-transfection of Flag-A51R vector. In (**A**) and (**B**) arrows indicate A51R filaments and arrowheads indicate A51R aggregate structures that overlap with tubulin staining. Note the absence of these aggregates and filaments in cells that lack Flag-A51R staining. Scale bars represent 10 µm. See also *Figure 7—figure supplements 1 and 2*.

The following figure supplements are available for figure 7:

**Figure supplement 1**. A51R localizes with and stabilizes MTs.

**Figure supplement 2**. Localization of Flag-A51R proteins encoded by VACV-MVA and disparate poxviruses in BSC-40 cells.

---

multidimensional protein identification technology (MuDPIT) (*Washburn et al., 2001*) to identify proteins associated with A51R immunoprecipitation (IP) complexes using mass spectrometry. We transfected LD652 cells with either empty vector or with Flag-A51R expression vector and then analyzed Flag antibody immunoprecipitates 48 hr post-transfection. We matched peptides identified in these LD652 cell immunoprecipitates to Flag-A51R sequence as well as to the available *Bombyx mori* proteome. Filtering differential normalized spectral abundance factor (NSAF) (*Zybailov et al., 2006*) values from both control and Flag-A51R immunoprecipitates identified the four most-enriched proteins in Flag-A51R treatments. These proteins are the target Flag-A51R, myosin II essential light chain (a constituent of the actomyosin cytoskeleton [*Clark et al., 2007*]), Ub, and ribosomal protein S15 (a component of the 40S subunit of the ribosome that has also been shown to modulate the activity of specific E3 ubiquitin ligases [*Daftuar et al., 2013*]) (*Table 2*). Unfortunately we found that the majority of A51R associated with insoluble cell fractions, making direct immunoblot analysis of immunoprecipitated materials impossible.

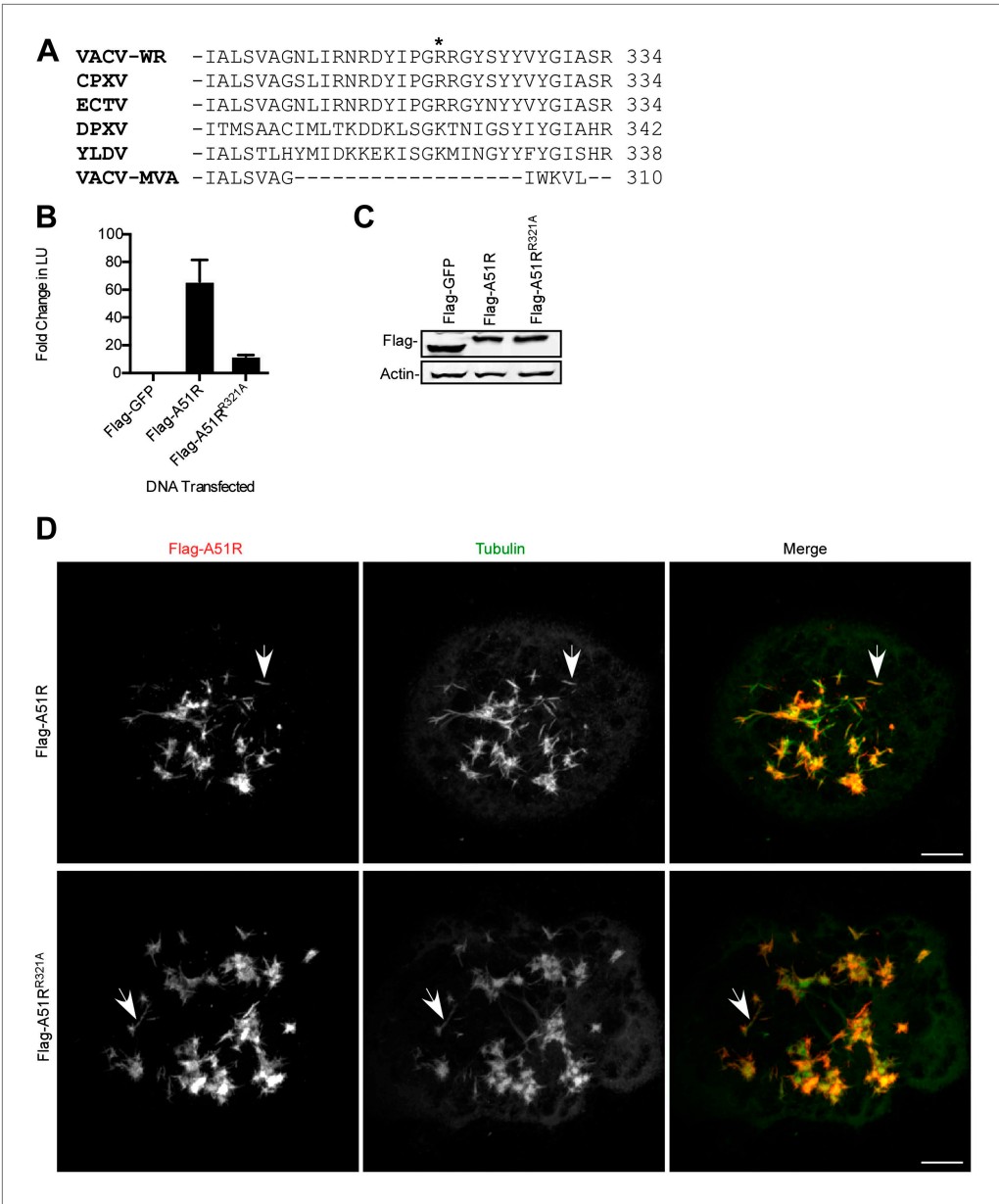

**Figure 8**. Microtubule stabilization is insufficient for full RNA virus rescue in LD652 cells. (**A**) C-terminal alignment of poxvirus A51R proteins with site of R321A substitution indicated by an asterisk. Multiple alignments were performed using eBioX Software (v. 1.5.1) using T-COFFEE alignment parameters. CPXV, cowpox virus; ECTV, ectromelia virus; DPXV, deerpox virus; and YLDV, Yaba-like disease virus. (**B**) Relative LUC activity in lysates of cells transfected with Flag-GFP, Flag-A51R or Flag-A51R^R321A p166 constructs for 24 hr and then infected with VSV-LUC for 48 hr. Fold change in LU are relative to LU obtained in Flag-GFP treatments. Data represent means (+SEM). (**C**) Immunoblot of Flag-GFP (lower band) and Flag-A51R/A51R^R321A (upper bands) from lysates in (**B**). (**D**) Immunofluorescence of Flag-A51R/A51R^R321A (red) and tubulin (green) proteins in LD652 cells in the presence of vincristine 48 hr post-transfection of Flag-A51R vectors. Arrows indicate A51R filaments that overlap with tubulin staining. Scale bars represent 10 μm.

Given our knowledge that RNAi of Ub transcripts (*Figure 2C,D*) or inhibition of the UPS system (*Figure 2—figure supplement 1*) enhances RNA virus gene expression in *L. dispar* cells and that A51R associates with Ub, we asked if A51R expression might affect viral protein stability. To first investigate if there were differences in viral protein translation rates in the presence of A51R, we used a two hour pulse of [$^{35}$S]methionine to label newly-synthesized proteins in virus-infected LD652 cells. We then analyzed the abundance of newly synthesized, radiolabeled VSV N protein after immunoprecipitation

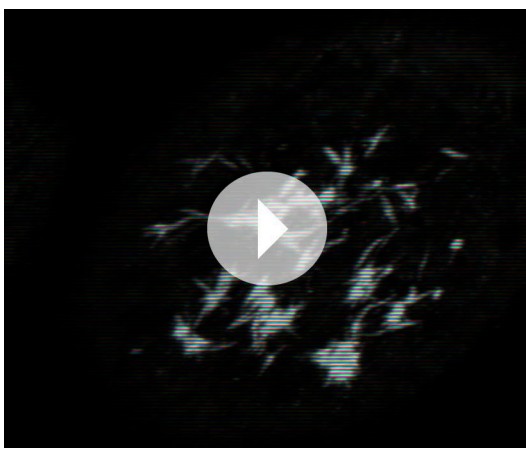

**Video 1**. Three-dimensional rendering of Flag-A51R-transfected LD652 cell in the presence of vincristine 48 hr post-transfection. A merge between Flag (red) and tubulin (green) staining is shown.

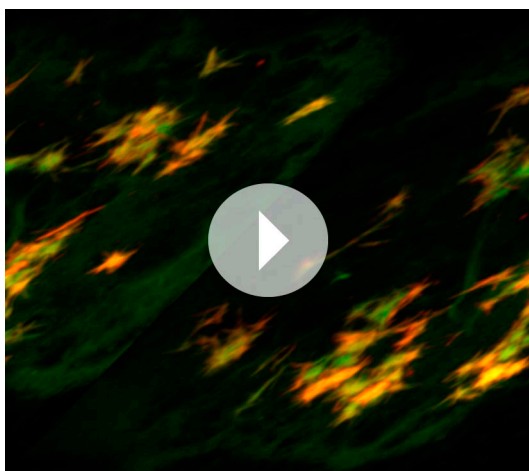

**Video 2**. Three-dimensional rendering of Flag-A51R<sup>R321A</sup>-transfected LD652 cell in the presence of vincristine 48 hr post-transfection. A merge between Flag (red) and tubulin (green) staining is shown.

**Table 2.** Proteins most-enriched in Flag-A51R immunoprecipitates from LD652 cells

| Protein | No. of spectral counts | ΔNSAF (NSAF$_{Flag-A51R}$−NSAF$_{Control}$) |
|---|---|---|
| Myosin II essential light chain | 110 | 0.0355 |
| Ub | 21 | 0.0113 |
| Flag-A51R | 33 | 0.0066 |
| Ribosomal protein S15 | 18 | 0.0056 |

from these lysates. We found that radiolabeled VSV N protein was undetectable in cells infected only with either VACV-WR (control) or VSV-LUC. In contrast, we found significant quantities of radiolabeled VSV N protein in VSV-LUC-infected cells that were also infected with either VACV-WR or ΔA51R strains (**Figure 9A**, top panel). These results suggest that viral protein translation rates are too low to be detected by this method in VSV-LUC single infections but are detectable (and surprisingly similar) in co-infections with VACV-WR or ΔA51R strains. When whole cell extracts from these experiments were immunoblotted for total VSV N protein levels, again we found that VSV N was not detected in either VACV-WR or VSV-LUC single infection lysates. However, total VSV N protein levels were dramatically reduced in ΔA51R co-infections compared to co-infections with VACV-WR (**Figure 9A**, middle panel). To determine if the reduced VSV N protein levels in ΔA51R co-infections might be due to altered VSV N protein stability, we calculated the half-life of newly-synthesized VSV N protein using pulse-chase analyses. Although newly synthesized VSV N protein could be detected during co-infection with either VACV-WR or ΔA51R strains (**Figure 9B**), the levels of radiolabeled protein appeared to diminish more rapidly during ΔA51R co-infection (**Figure 9C**). Regression analyses of pulse-chase experiments estimated the half-life of VSV N to be ~5.3 hr during co-infection with VACV-WR while only ~1.8 hr during co-infection with the ΔA51R strain. Collectively, these data suggest that A51R associates with Ub and promotes viral protein stability.

## A51R promotes VACV replication and pathogenesis in vertebrates

Given that A51R is encoded by vertebrate poxviruses, we wished to determine if A51R contributes to VACV replication and pathogenesis in vertebrate hosts. To do this we first performed growth curve analyses after infection of BSC-40 cells. At a low MOI of 0.03, the ΔA51R, VACV-WR, and ΔA51R<sup>FA51R</sup> strains replicated with similar kinetics until 24 hpi at which point ΔA51R titers began to plateau. In contrast, VACV-WR and ΔA51R<sup>FA51R</sup> titers continued to increase to ~100-fold higher levels by 72 hpi (**Figure 10A**). This plateau of ΔA51R titer was also observed at a higher MOI of 3, but occurred earlier, at 12 hpi (**Figure 10B**). These results indicate that A51R promotes VACV replication in vertebrate cell culture.

To determine if A51R also promotes VACV pathogenesis, we intranasally-infected groups of NMRI mice with either VACV-WR or ΔA51R and

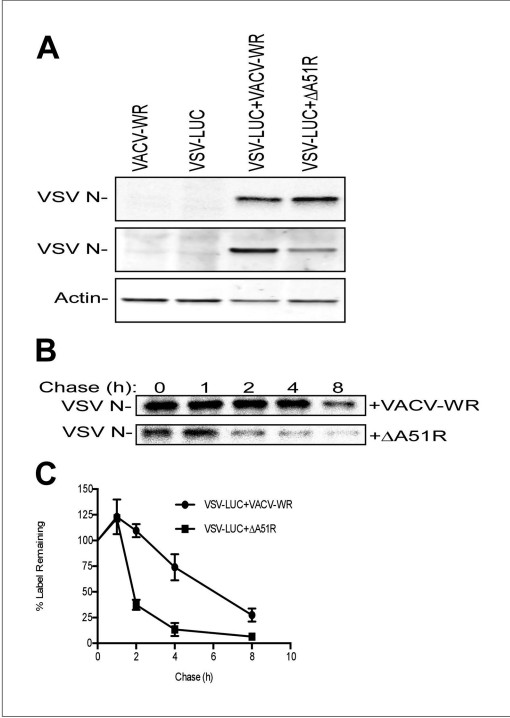

**Figure 9**. A51R promotes RNA virus protein stability in LD652 cells. Cells were infected with the indicated strains for 72 hr and then pulsed for 2 hr with [³⁵S]methionine. To visualize radiolabeled, nascent VSV N protein, cell lysates were subjected to immunoprecpitation with anti-VSV N protein antibodies and immunoprecipitated complexes were separated by SDS-PAGE and visualized by autoradiography (top panel). Equal fractions of each total cellular lysate were also used in parallel immunoblot experiments to detect total VSV N (middle panel) or actin protein levels. (**B** and **C**) Cells were infected with VSV-LUC and the indicated VACV strains as in (**A**), pulsed for 2 hr with [³⁵S]methionine and then chased for the indicated times at which point cell lysates were collected and subjected to immunoprecipitation with anti-VSV N antibodies. Immunoprecipitated complexes were then separated by SDS-PAGE, visualized by autoradiography (**B**) and the percentage of radiolabeled VSV N protein remaining (compared to T = 0, representing the end point of the 2 hr pulse) at each time point post-pulse was plotted (**C**). Note that lysates containing radiolabeled VSV N protein from VACV-WR and ΔA51R co-infections time courses (**B**) were processed separately and thus band intensities between VACV infection treatments are not directly comparable. Data in (**C**) represent means (+SD).

then tracked body weight (*Figure 10C*) and survival (*Figure 10D*) of animals over time. Both VACV-WR and ΔA51R infections led to rapid weight loss of animals by days 4 and 5, respectively. In animals infected with VACV-WR, weight loss continued on days 6–7 post-infection, with 5/5 animals ultimately succumbing to infection by day 7. In contrast, only 1/5 animals infected with the ΔA51R strain died by day 7, while the remaining animals recovered from infection, gained weight (*Figure 10C*) and survived until the end of the 28 day experiment (*Figure 10D*).

We further confirmed in separate experiments, and at multiple doses of virus, that the ΔA51R virus displays attenuated virulence compared to VACV-WR (*Figure 10—figure supplement 1A,B*). To determine the relative amount of virus replication that occurred during these infections in vivo, we isolated lung and ovarian tissues on days 4 and 6 post-infection and measured virus titers by plaque assay. After four days of infection, both VACV-WR and ΔA51R viruses replicated to similar titers in lung tissue, the primary site of infection. By day 6, however, a trend toward lower ΔA51R titers in the lungs was observed, although this difference was not statistically significant (p=0.05; *Figure 10E*). Neither VACV-WR nor ΔA51R viruses were detected in isolated ovaries by day 4 post-infection, suggesting that these viruses had not yet spread to this distant tissue. By day 6 post-infection, however, we detected significantly higher virus titers (p<0.05) in the ovaries of all VACV-WR-infected animals compared to ΔA51R-infected animals, in which virus was detectable in only 1/5 animals (*Figure 10F*). These data suggest that A51R is not only important for VACV replication in cell culture but also for virulence in vivo.

The reduced replication of the ΔA51R strain at later time points of infection during in vitro and in vivo infections was not caused by a reduction in the extracellular enveloped (EEV) form of VACV, which primarily mediates spread of the virus (*Smith et al., 2002*). Our analysis revealed that, despite ~10-fold lower titers of total (intracellular and extracellular) virus in ΔA51R cultures than in VACV-WR or ΔA51R^FA51R cultures, ΔA51R cultures shed ~twofold higher levels of EEV than VACV-WR or ΔA51R^FA51R cultures (*Figure 10—figure supplement 1C*).

The early plateau of ΔA51R replication and reduced load of this mutant in ovarian tissues might be explained by an inability to overcome an immune response caused by initial infection. We therefore pre-treated BSC-40 cells with interferon (IFN) to induce an antiviral state in cells prior to VACV infection and asked whether A51R is required to overcome an IFN response. Comparing the titers of VACV-WR, ΔA51R, or ΔA51R^FA51R cultures pre-treated with IFN to parallel cultures without IFN treatment, we found that IFN pre-treatment reduced VACV-WR and ΔA51R^FA51R titers by ~10-fold by 24 hpi relative to control treatment. ΔA51R titers,

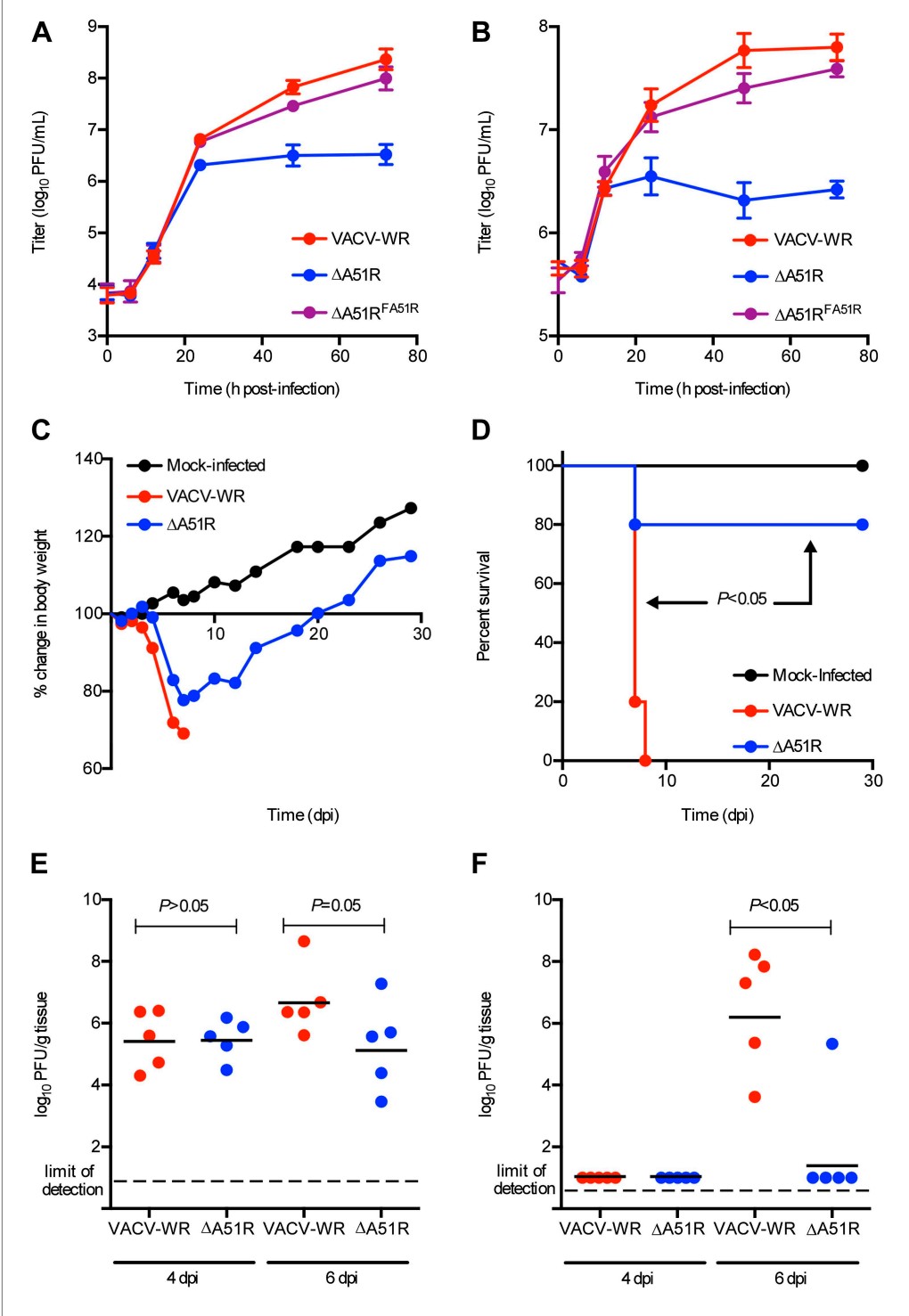

**Figure 10**. A51R promotes VACV replication and pathogenesis in vertebrates. (**A** and **B**) VACV titers from BSC-40 cells infected with the indicated strains at low MOI (0.03) (**A**) or high MOI (3) (**B**). Data represent means (+/-SEM). (**C**) Body weight of NMRI mice infected with the indicated virus (10,000 PFU/animal). Data represent mean percent body weight change among surviving members in each group at the indicated day post-infection. (**D**) Percent survival of mice from (**C**). p<0.05 indicates a statistically significant difference in survival between VACV-WR- and ΔA51R strain-infected animals. (**E** and **F**) Virus titer in lung (**F**) and ovary (**F**) tissue from mice infected with VACV-WR or ΔA51R. Each dot indicates the total virus titers from an individual mouse. Horizontal bars represent the mean of

*Figure 10. Continued on next page*

*Figure 10. Continued*

each group. Mice were infected as in (**C**) and euthanized at 4 or 6 days post-infection. p<0.05 indicates a statistically significant difference between infection group tissue titers. See also *Figure 10—figure supplement 1*.

The following figure supplements are available for figure 10:

**Figure supplement 1**. The ΔA51R strain displays attenuated pathogenesis in NMRI mice, releases increased levels of EEV, and does not display enhanced sensitivity to IFN treatment.

however, were similarly reduced at this time point, suggesting that loss of A51R does not make VACV more susceptible to the antiviral effects of IFN (*Figure 10—figure supplement 1D*).

## Discussion

Insect model systems provide powerful tools for probing virus-host interactions but have primarily focused on Dipteran hosts (*Kingsolver et al., 2013*). Development of new virus-host models in *Lepidoptera* is important for several reasons. First, these insects may encode antiviral mechanisms that are not found in *Diptera*. Second, our current understanding of Lepidopteran antiviral immunity is solely garnered from studies of invertebrate DNA viruses, leaving many questions regarding how these organisms restrict RNA virus infection. Third, understanding how Lepidopterans combat virus infection may lead to more rationale design of virus-based 'bioinsecticides' to control pest species.

Here we have identified an unusual resistance of Lepidopteran cells to VSV and SINV infection, and in so doing, we have further developed LD652 cells as a useful tool for the study of virus-host interplay in an important pest species. Our mRNA-seq-based transcriptome data, combined with RNAi knock-downs studies, allowed us to identify host cell immunity factors that restrict RNA virus replication. We also used an RNAi screen to identify A51R as a poxvirus factor that overcomes Lepidopteran immunity to virus infection. A51R stabilizes a subset of host MTs and promotes RNA virus protein stability, which might be explained by its association with Ub. In addition we have shown that A51R is critical for VACV infection in vertebrate cells and is a virulence factor in mice. Our findings raise the intriguing possibility that A51R promotes both DNA and RNA virus infection by disarming host immunity mechanisms conserved in both vertebrates and invertebrates.

Despite promiscuous host ranges in both vertebrates and invertebrates, VSV and SINV were completely restricted in LD652 cells. Restriction occurred post-entry, and was overcome either by VACV-co-infection or by blocking host cell transcription. Using RNAi we have identified several host-cell factors that restrict RNA viruses in Lepidopteran cells. These include; (i) homologs of the antiviral RNAi factors AGO2 and Dicer-2 (*Kingsolver et al., 2013*), (ii) Relish, a homolog of vertebrate NF-κB family members of antimicrobial transcription factors (*Dushay et al., 1996*), (iii) a homolog of 'Effete' an E2 Ub-conjugating enzyme (*Treier et al., 1992*), and finally (iv) several genes encoding Ub.

A role for AGO2 and Dicer-2 in Lepidopteran restriction to RNA virus replication is consistent with the deep conservation of RNAi-based antiviral immunity in vertebrates and invertebrates (*Kingsolver et al., 2013*; *Li et al., 2013*; *Maillard et al., 2013*). Indeed, transient expression of the nodamura virus B2 protein, a well-characterized inhibitor of both invertebrate and vertebrate RNAi pathways (*Sullivan and Ganem, 2005*), also rescues VSV and SINV replication in multiple Lepidopteran cell lines (DBG and CCM, unpublished data). Thus, Lepidopteran cells provide a new system to investigate the mechanism of antiviral RNAi. These studies will likely complement other RNA virus-invertebrate host models such as the Orsay virus-*Caenorhabditis elegans* system, in which RNAi has been shown to restrict RNA virus replication (*Felix et al., 2011*). We are currently conducting deep-sequencing of small RNAs to further explore how Lepidopteran RNAi pathways respond to RNA virus infection.

In *Drosophila*, the NF-κB homolog Relish is a component of the IMD pathway which is stimulated by receptor-dependent recognition of bacterial peptidoglycan (*Kleino and Silverman, 2014*). Pathway stimulation results in phosphorylation and cleavage of Relish, which then translocates to the nucleus to activate transcription of antimicrobial factors (*Silverman et al., 2000*). Activation of Relish requires a series of ubiquitination reactions that are dependent on a complex of Ub-conjugating enzymes that includes Effete (*Paquette et al., 2010*). The finding that homologs of Relish, Effete and Ub all function to restrict VSV and SINV replication in *L. dispar* cells suggests that an IMD-like pathway protects against virus infection in *Lepidoptera*. Our findings are consistent with recent reports that the IMD

pathway plays an antiviral role in *Diptera* (*Avadhanula et al., 2009*; *Costa et al., 2009*). However, the mechanism by which viruses are recognized by this pathway is still unknown.

While it is tempting to speculate that the Ub-encoding transcripts identified here promote viral immunity via Relish activation, Ub has a role in many cellular processes and therefore may be required for a variety of immune functions (*Oudshoorn et al., 2012*). Interestingly, several natural viral pathogens of *Lepidoptera* encode Ub-like molecules, suggesting that Ub may be required for their life cycle (*Barry et al., 2010*). On the other hand, viruses may also benefit from inhibiting or reversing ubiquitination. For example, baculovirus-encoded Ub homologs can inhibit Ub chain elongation (*Haas et al., 1996*), and several viruses encode potent deubiquitinating enzymes (*Lindner, 2007*). Inhibiting or reversing Ub reactions might block the activation of the IMD pathway and/or may prevent the direct targeting of viral proteins for destruction by the host UPS. Future studies will be aimed at dissecting the roles of these *L. dispar* host factors in restricting virus replication.

We were surprised by the rescue of VSV and SINV replication in LD652 cells by VACV, despite the fact that VACV replication is abortive in these cells (*Li et al., 1998*). To our knowledge, this is the first example of heterologous virus rescue by a vertebrate virus in an invertebrate host. In mammalian cell culture, VACV also enhances VSV replication, but the VACV-encoded IFN response antagonists E3L and B18R are the major determinants of this rescue (*Le Boeuf et al., 2010*; *Shors et al., 1998*). Since the IFN response appears to be vertebrate-specific (*Mukherjee et al., 2014*), it was perhaps not surprising that we failed to observe any effect of *E3L* or *B18R* deletion on VACV-dependent VSV rescue in *L. dispar* cells (data not shown). Instead, we identified the previously uncharacterized VACV gene product A51R as a major determinant of VACV-mediated RNA virus rescue. Additional VACV proteins are likely to contribute to suppression of antiviral activity in *L. dispar* cells, since both VSV and SINV replication were partially rescued by co-infection with the ΔA51R strain. This might explain why newly synthesized, radiolableled VSV N proteins were undetectable in single infections yet at relatively abundant (and similar) levels during co-infection with either VACV-WR or ΔA51R strains, despite the lack of N protein accumulation in the latter co-infection. Thus, the highly restrictive nature of VSV and SINV infections in *L. dispar* cells, in combination with our sensitive LUC-based detection methods, may prove useful in identifying additional novel immunomodulators encoded by VACV and other viruses.

Although our understanding of how A51R promotes RNA virus replication is incomplete there are several intriguing clues. First, our analysis indicates that A51R-mediated rescue occurs at a step after RNA virus entry. Second, the findings that A51R promotes RNA virus infection in multiple Lepidopteran (but not Dipteran) cells and promotes VACV infection in vertebrate hosts, suggests that A51R overcomes immunity mechanisms that are shared between Lepidopterans and mammals but absent or easily antagonized in Dipteran insects. Third, the suppression of host restriction to RNA virus replication and the association with MTs are both conserved features of A51R homologs from disparate vertebrate poxviruses, suggesting that both immunomodulatory and MT-stabilization functions may be important for poxvirus replication. This idea is supported by the reduced replication of the ΔA51R strain in vertebrate cell culture and in mice. Finally, the association of A51R with Ub and the A51R-dependent VSV protein stabilization we observed, along with the aforementioned roles of the UPS in restricting RNA virus replication, point to a role for A51R in usurping or inhibiting host Ub machinery.

A51R proteins localize with a subset of host MTs in both invertebrate and vertebrate cells, often forming tubulin aggregates and filament-like structures that are resistant to MT depolymerizing agents. These findings suggest a close interaction of A51R with MT structures, and consistent with this, A51R co-sedimented with MTs in MT-pelleting assays (DBG and CCM unpublished data) which may explain our difficulty in recovering soluble A51R in immunoprecipitation studies. Importantly, these nocodazole-resistant MT structures are absent in ΔA51R infections and can be formed by expression of A51R in the absence of other poxvirus proteins. Furthermore, the truncated VACV-MVA A51R protein fails to form filamentous, nocodazole-resistant MTs. Thus, the presence or absence of functional A51R could explain why the nocodazole-resistant 'MT pieces' previously observed in VACV-infected cells were not observed in uninfected or VACV-MVA-infected cells (*Ploubidou et al., 2000*; *Schepis et al., 2006*). Our results suggest that A51R is the elusive MT-stabilizing factor encoded by VACV.

When we initially observed an inability of VACV-MVA A51R to both rescue RNA virus replication and stabilize MTs, we hypothesized that stabilization of MTs may be the mechanism by which A51R promotes virus replication. Previous studies implicate MTs in promoting VSV and SINV gene expression (*Qiu et al., 1998*; *Heinrich et al., 2010*), and tubulin has been shown to promote VSV RNA polymerase activity in vitro (*Moyer et al., 1986*). Furthermore, VSV transport to the cell surface is reported to be

MT-dependent (*Das et al., 2006*). However, a recent study found no effect of MT depolymerization on VSV and SINV replication (*Matthews et al., 2013*). Thus the role of MTs in VSV and SINV replication remains unresolved. Considering the very early restriction of RNA virus gene expression in LD652 cells, it seems unlikely that A51R overcomes host restriction by promoting RNA virus transport on stabilized MTs. Furthermore, our studies with the A51R$^{R321A}$ mutant indicate that RNA virus rescue is not fully mediated by MT association or stabilization and therefore must require an additional function encoded by A51R. This finding however, does not rule out the possibility that MT-association is required for A51R immunosuppressive function or stability.

It is possible that A51R subverts host antiviral responses partly by perturbing MT-dependent transport of host factors and/or organelles. Previously it has been shown that VACV-WR infection causes redistribution of endosomes from dispersed cytosolic locations to large perinuclear clusters, suggesting that outward, MT-dependent endosomal trafficking might be impaired during VACV infection (*Schepis et al., 2006*). This endosome clustering phenomenon requires early gene expression and one or more activities lacking in VACV-MVA (*Schepis et al., 2006*). A51R fits these criteria, and thus could be the VACV factor that induces endosome clustering, perhaps to overcome a host mechanism that could otherwise trap viruses in endosomes. It is interesting to note that even at relatively late times post-infection (e.g., 24 hr) we detected punctate immunofluorescence staining patterns for VSV N and SINV E1 virion-associated proteins inside LD652 cells. It is possible that these punctate structures represent virions trapped inside endosomes as both VSV (*Mire et al., 2010*) and SINV (*DeTulleo and Kirchhausen, 1998*) can utilize endocytic modes of entry. It will be interesting in the future to determine if the expression of wild-type or mutant forms of A51R alter endosomal distribution in *L. dispar* cells and whether endosomal redistribution correlates with virus replication. Unfortunately, we have not yet found suitable reagents for monitoring endosome distributions in *L. dispar*.

Alteration of MT-dependent trafficking by A51R may help explain why both VACV-MVA (*Meiser et al., 2003*) and ΔA51R (this study) strains release 2-3-times more EEV particles than does VACV-WR. This phenomenon is particularly striking given the overall 10-fold lower total (intracellular + extracellular) virus titers of the ΔA51R strain. The release of EEV is MT-dependent (*Ploubidou et al., 2000*), and EEV is known to elicit neutralizing antibodies (*Smith et al., 2002*). Thus it is possible that A51R attenuates virion transport along MTs, and subsequent EEV release, in order to allow the virus to escape activation of host immune responses (*Schepis et al., 2006*). Interestingly, the Dryvax strain of VACV, which is used for smallpox vaccination, encodes a fragmented *A51R* gene (*Qin et al., 2011*). These observations, along with the attenuated virulence of ΔA51R strain in vivo, raise the intriguing possibility that loss of A51R function contributes to the induction of poxvirus immunity.

It is also possible that MT structures might serve as a scaffold to allow A51R to interact with and sequester host factors, such as Ub. Although we do not yet know specifically how A51R promotes virus protein stability, it is possible that A51R-dependent sequestration of Ub (or removal of Ub from viral proteins) acts to overcome a Ub-dependent mechanism that either activates host antiviral protein effectors or directly targets nascent viral proteins for destruction. This latter hypothesis predicts that Lepidopterans cells might have a mechanism to target protein synthesis that is coupled to cytoplasmic transcription. Importantly, inhibition of the UPS in *L. dispar* cells only enhanced LUC activity when cells were infected with LUC-encoding viruses and not when cells were simply transfected with a LUC expression plasmid. Given that VACV, VSV and SINV are all cytoplasmically-replicating viruses and that plasmid-based transcription of *luc* mRNA would occur in the nucleus, this hypothesis is not unreasonable. Whether other potential A51R interactors identified here such as myosin II essential light chain or RPS15 play a role in A51R rescue function is currently unknown.

Further studies of A51R function will not only help us to understand eukaryotic antiviral immunity but may also provide new strategies to overcome the antiviral responses of Lepidopteran pests, leading to more effective biocontrol agents. Indeed, the unexpected uncoupling of RNA virus restriction by A51R has provided an exciting new arena to explore virus-host interactions.

## Materials and methods

### Chemicals

X-gal, nocodazole, vincristine, PYR-41, and MG132 were obtained from Sigma-Aldrich (St. Louis, Mo) and dissolved in dimethylsulfoxide (DMSO). AraC (Sigma) was dissolved in sterile water. Recombinant IFN-α solution was from Sigma. X-glu (Clontech, Palo Alto, CA) was dissolved in DMSO. Compounds

were diluted to their final concentration in cell culture medium or in a 1:1 mixture of 2 × Dulbecco's minimal essential medium (DMEM; Invitrogen, Carlsbad, CA) and 1.7% agar (X-gal; X-glu) immediately prior to use.

## Cell and virus culture

African Green Monkey kidney cells (BSC-40), baby hamster kidney (BHK), and human embryonic lung (HEL) cells were obtained from American Type Culture Collection (ATCC). BSC-40 and HEL cells were maintained in MEM containing 10% Fetal bovine serum (FBS). BHK cells were maintained in DMEM containing 10% FBS. *L. dispar*-derived LD652 cells were obtained from Dr Basil Arif (Natural Resources Canada, Canada) and were maintained in a 1:1 mixture of Ex-Cell 420 (Sigma) and Graces insect medium (Invitrogen) that also contained 10% FBS. *L.dispar*-derived embryonic (LdEP) cells were obtained from the United States Department of Agriculture Research Service and were maintained in Ex-Cell 420 medium supplemented with 3% FBS. *S. frugiperda*-derived Sf9 cells were obtained from Invitrogen and were maintained in Sf-900 II medium (Invitrogen) supplemented with 10% FBS. *M. sexta*-derived GV-1 cells were obtained from Dr Que Lan (University of Wisconsin) and were maintained in Graces insect medium supplemented with 10% FBS. *D. melanogaster*-derived DL1 cells were obtained from Dr Sara Cherry (University of Pennsylvania) and were maintained in S2 medium (Invitrogen) containing 10% FBS. All medium also contained 1% antibiotic-antimycotic (Invitrogen). Vertebrate cells were incubated at 37°C in a 5% $CO_2$ atmosphere and invertebrate cells were incubated at 27°C in normal atmosphere.

VACV-WR, VACV-COP, and VACV-MVA were obtained from Dr Anuja Mathew (University of Massachusetts Medical School). The A5L-GFP VACV strain (*Carter et al., 2003*) was obtained from Dr Geoff Smith (University of Cambridge, United Kingdom). VSV-GFP (*Kato et al., 2005*) was obtained from Dr Hiroki Kato (Kyoto University, Japan). VSV-LUC (rVSVluc; *Cureton et al., 2009*) was obtained from Dr Sean Whelan (Harvard Medical School). SINV-GFP (TE/5'2J/GFP-3XFlag; *Cristea et al., 2006*) was obtained from Dr Margaret MacDonald (Rockefeller University) and SINV-LUC [TRNSVluc; *Cook and Griffin, 2003*] was obtained from Dr Dianne Griffin (Johns Hopkins University).

VACV stocks were amplified in BSC-40 cells with the exception of VACV-MVA which was passaged in BHK cells. VACV strains were titrated on BSC-40 cells by plaque assay as previously described (*Gammon et al., 2010*). VSV and SINV were amplified using low MOI infections in BHK cells. Viruses were collected from culture supernatants by ultracentrifugation (25,000 rpm, 2 hr, 4°C) and titrated by plaque assay on BSC-40 cell monolayers overlayed with a 1.5% low-melting point agarose (Invitrogen). All MOIs reported for vertebrate viruses (VACV, VSV, SINV) refer to those calculated from mammalian cell-based plaque assays.

Virus infections were carried out for 2 hr in either serum-free DMEM at 37°C (vertebrate cell infections) or in Sf-900 II serum free media at 27°C (invertebrate cell infections). After 2 hr, inocula were replaced with normal growth medium. When titering VSV/SINV from cultures co-infected with VACV, AraC (50 µg/ml) was included in the agarose overlays. Where indicated, AraC was added to cultures at the indicated doses 2 hpi in the appropriate growth medium or VACV virions were heat-inactivated prior to infection by incubation at 55°C for 1.5 hr (*Dales and Kajioka, 1964*). Unless otherwise noted, invertebrate cells were infected at MOIs of 100 (VACV) or 10 (VSV/SINV). At these MOIs ~100% of cells are infected with VACV by 48 hpi (*Li et al., 1998*).

## Recombinant virus construction

### General recombinant virus methods

Recombinant viruses were generated using previously described methods (*Gammon et al., 2010*). Briefly, BSC-40 cells were grown to confluence in 6-well dishes and then infected for 1 hr with the appropriate VACV strain (see below) at a MOI of 2 in 0.5 ml of PBS. The cells were then transfected with 2 µg of linearized plasmid DNA using Lipofectamine 2000 (Invitrogen) in Opt-MEM media (Invitrogen). The cells were returned to the incubator for another 5 hr, the transfection solution was replaced with 2 ml of fresh growth medium, and the cells were cultured for 24–48 hr at 37°C. Virus progeny were released by freeze-thawing, and the virus titer was determined on BSC-40 cells. To identify recombinant virus, plaques were stained with X-gal or X-glu (both at 0.4 mg/ml) in solid growth media. PCR was used to confirm insertions/deletions in the resulting recombinant viruses.

### ΔA51R, ΔA51R^FREV, ΔA51R/ΔJ2R, ΔA51R/ΔJ2R^FA51R, and ΔA51R/ΔJ2R^FMVAA51R strain construction

The plasmid pZIPPY-NEO/GUS (*Dvoracek and Shors, 2003*) was used to clone an ~500 bp PCR product (using primers: 5'- ACTAGTCGAACCGGGAAAGAGAAGAT-3' & 5'-AAGCTTGTATGTAACTATTAA

GATTT-3') containing sequences flanking the 'A50R' side of the A51R locus as well as an ~520 bp PCR product (using primers: 5'-CCGCGGATTAAGATTGCTCTTTCGGT-3' & 5'-AGATCTTAAAGTTATCT GCTCCCTCT-3') containing sequences flanking the 'A52R' side of the A51R locus. The 500 bp PCR fragment was cloned into pZIPPY-NEO/GUS using SpeI and HindIII restriction sites and the 520 bp PCR fragment was cloned into the resulting vector using SacII and BglII restriction sites. Rescue of this vector (now called pZIPPY-A50R^H+A52R^H) leads to the deletion of nts 157628-158568 in the VACV-WR genome (GenBank accession: NC_006998). The deleted region is replaced by a p7.5-promoted neomycin resistance (neo) gene as well as a bacterial gusA gene under the control of a modified H5 promoter. To generate the ΔA51R strain, pZIPPY-A50R^H+A52R^H DNA was transfected into cells infected with VACV-WR and recombinant viruses were selected by blue coloration in the presence of X-glu.

A ΔA51R revertant strain (ΔA51R^FREV) was constructed by rescue of a plasmid (synthesized by Invitrogen) encoding an N-terminal Flag-tagged A51R gene flanked on either side by ~500 bp of VACV-WR sequence. Rescue of this plasmid (pA51RFREV) into the ΔA51R background introduces the Flag-A51R gene into the A51R locus, placing it under the natural A51R promoter. Selecting clear plaques in the presence of X-glu led to the isolation of these revertant viruses.

To generate the ΔA51R/ΔJ2R strain, the previously described transfer vector, pSC66 (**Gammon et al., 2010**) was used to insertionally-inactivate the J2R locus. This pSC66 vector was also used to introduce a codon-optimized (see below), Flag-A51R gene into the J2R locus of the ΔA51R strain, creating the ΔA51R/ΔJ2R^FA51R strain. Similarly, pSC66 was also used to transfer a codon-optimized (see below), Flag-A51R gene from VACV-MVA into the J2R locus, generating the ΔA51R/ΔJ2R^FMVAA51R strain. Recombinant viruses were selected in the presence of X-gal as previously described (**Gammon et al., 2010**). All virus strains were plaque-purified a minimum of three times in BSC-40 cells and were analyzed by PCR and sequence analysis to confirm deletion/insertions (data not shown).

## RNA isolation and RT-PCR

Total RNA was collected from cell cultures in TRIzol Reagent (Life Technologies). The aqueous layer was isolated using phase lock columns (5 Prime Gaithersburg, MD) and RNA was isopropanol-precipitated and resuspended in nuclease-free water. Genomic DNA was removed using a Turbo DNA-free kit (Invitrogen). First-strand cDNA was synthesized using 0.5–1 µg of purified total RNA, gene-specific primers, and Superscript III Reverse Transcriptase (Invitrogen), according to the manufacturer's recommendations. PCR was performed with Takara Ex-Taq (Clontech, Mountain View, CA). PCR cycling conditions were: 94°C for 3 min, followed by 94°C for 30 s, 50°C for 30 s, 72°C for 1 min for 36 cycles. PCR products were separated on 2% agarose gels containing ethidium bromide and images were captured using an ECII Darkroom System with Labworks acquisition software (version 4.0.0.8; Bioimaging Systems, Upland, CA). VSV (+)-sense transcription was analyzed by RT-PCR using primers: 5'-ATGTCTA CAGAAGATGTA-3' & 5'-TAATATATAATAGGTGATCTGAGAATTATAGGGTC-3' (**Wilkins et al., 2005**). SINV (−)-sense transcript levels were analyzed by RT-PCR using primers: 5'-TAGACAGAACTGA CGCGGACGT-3' & 5'-TCCATACTAACTCATCGTCGATCTC-3' (**Campbell et al., 2008**). RT-PCR amplification of L. dispar actin transcripts was performed with primers: 5'-GGGACAGAAGGACTCGTACG-3' & 5'-GCCTTAGGGTTGAGAGGAGC-3' (**Chen et al., 2003**).

## mRNA-sequencing

Total RNA from LD652 cells was submitted to the Georgia Genomics Facility (Athens, GA) for library preparation and SE100 sequencing. TruSeq RNA libraries were prepared from total RNAs and rRNA depletion was performed. SE100 reads were sequenced from resulting libraries using an Illumina HiSeq 1000 instrument. This sample yielded 238,502,918 raw reads, which were subjected to quality control procedures implemented using the FASTX-toolkit (Provided by Dr G Hannon, Cold Spring Harbor Laboratories, NY). Specifically, artifact reads were eliminated, and end sequences whose Phred scores corresponded to an error rate exceeding 1% were clipped. Only end-trimmed reads of 36 bases or longer were retained, and these were also required to have not less than 90% of their bases possessing a Phred score of 21 or higher. Post cleaning, the sample had 122,295,683 sequences (12,148,224,605 bases). These were assembled into 115,739 putatively unique transcripts (PUTs) using the Trinity assembly program (**Grabherr et al., 2011**) and compared with the 4 September 2013 version of NCBI NR using Blastx (**Altschul et al., 1997**). PUTs were partitioned into gold (7055 entries), silver (6353) and bronze (8251) tiers on the basis of alignment quality as described in **Sparks et al.**

*(2013)*. The gold and silver tiers collectively defined a set of 13,408 unique PUTs mapping to 7593 unique NCBI NR records. These gold and silver tier PUTs were used to identify potential immunity-related *L. dispar* transcripts and for the design of dsRNAs for RNAi. The cDNA sequences of the *L. dispar* transcripts identified in *Figure 2C,D* are available in *Supplementary file 1*.

## RNAi knockdowns, transfections and LUC assays

dsRNAs (~400 bp in length) were transcribed in vitro using the Megascript RNAi kit (Life Technologies). Templates were generated by RT-PCR reactions using gene-specific primers tailed at the 5′ end with the T7 promoter sequence (TAATACGACTCACTATAGGG). Sequences of primers used to make dsRNA targeting *L. dispar* mRNAs are listed, along with target sequences in *Supplementary file 1*. Sequences of primers used to generate dsRNA against VACV targets are listed in *Supplementary file 2*. dsRNA (~1 μg) was transfected into $10^5$ LD652 cells using Cellfectin II (Invitrogen) in Sf-900 II media according to the manufacturer's guidelines. DNA transfections (~1 μg) were also performed using Cellfectin II in Sf-900 II media. For VACV RNAi, after 5 hr the transfection media was replaced with virus inocula for 2 hr and then with regular growth medium. For *L. dispar* RNAi, after 5 hr the transfection medium was replaced with regular growth medium. Transfected cells were then challenged with VSV or SINV 24 hr post-transfection. RNAi transfection conditions resulted in >80% knockdown of either VACV or host transcripts (data not shown).

At the indicated times post-infection/transfection, cells were briefly washed in phosphate buffered saline (PBS) collected by centrifugation (2000 rpm, 10 min, 4°C) and lysed in reporter lysis buffer (Promega, Madison, WI) according to the manufacturer's guidelines. Lysates were spotted to 96-well dishes, mixed with Luciferase Assay Reagent (Promega) and arbitrary light units (LU) were measured using an Envision 2102 Multilabel Reader with Wallac EnVision Manager software (version 1.12; PerkinElmer, Waltham, MA). All experiments were performed at least three times.

## Expression constructs

For PCR amplifications for use in downstream cloning and gene expression studies, *iproof* DNA polymerase (Bio-Rad, Hercules, CA) was used. All final expression constructs were confirmed by DNA sequencing. Cellfectin II and Lipofectamine 2000 were used according to the manufacturer's guidelines for transfection of invertebrate and vertebrate cells, respectively. Invertebrate cell transfections were performed in SF-900 II medium and vertebrate cell transfections were performed in Opti-MEM for 5–6 hr after which the transfection medium was replaced with normal growth media. Where indicated, normal growth media containing nocodazole (20 μM) or vincristine (4 μM) was used to replace transfection media.

To express genes in invertebrate cells, either a modified p166 vector (*Lin et al., 2001*) or the pIZ/V5-His vector (Invitrogen) was used for transient expression. To facilitate cloning, the p166 vector was modified by introduction of a new multiple cloning site (MCS) between the *Bam*HI and *Xba*I sites using linker ligation and primers: 5′-GATCCCCCGGGACCGCGGCAAGTCGACCAATCGCGAAAGGAATTC AAGTTAATTAAT-3′ & 5′-CTAGATTAATTAACTTGAATTCCTTTCGCGATTGGTCGACTTGCCGCGG TCCCGGGG-3′. A Flag-tagged *gfp* gene was amplified from pEGFP-C3 vector (Clontech) DNA using primers: 5′-CCGCGGATGGATTATAAGGATGATGATGATAAGATGGTG-3′ & 5′-TTAATTAATTAC TTGTACAGCTCGTCCATGCC-3′ and cloned into the *Sac*II/*Pac*I sites of p166. A plasmid encoding a codon-optimized, Flag-tagged VACV-WR *A51R* gene was synthesized by Invitrogen and used for cloning into the p166 vector using *Sac*II/*Pac*I sites. Plasmids encoding codon-optimized, Flag-tagged *A51R* genes from CPXV (strain Brighton), ECTV (strain Moscow), DPXV (strain W-1170-84), YLDV (strain Davis), and MYXV (strain Lausanne) were also synthesized and these genes were digested from these initial vectors and ligated to *Sac*II/*Pac*I-cut p166. To express Flag-A51R, and Flag-tagged forms of CPXV/ECTV/DPXV/YLDV/MYXV A51R proteins in vertebrate cells, each p166 vector was *Sac*II/*Pac*I digested and inserts were cloned into pCDNA3 (Invitrogen) that had been modified to encode the same MCS as the p166 vector using linker ligation and primers described above. A QuikChange II Site-Directed Mutagenesis Kit (Agilent Technologies, Santa Clara, CA) and primers: 5′-GTAACCGTGA CTACATCCCCGGAGCTCGTGGTTACTCCTACTAC-3′ & 5′-GTAGTAGGAGTAACCACGAGCTCCGGGG ATGTAGTCACGGTTAC-3′ were used to generate a VACV *Flag-A51R* p166 vector encoding the R321A amino acid substitution.

To express the Severe Acute Respiratory Syndrome Coronavirus PLpro deubiquinase and its catalytically-inactive mutant form (PLpro[C112A]) (*Barretto et al., 2005*) in invertebrate cells, pCDNA3 vectors

encoding the wild-type and mutant forms (kind gifts from Dr Susan Baker, Loyola University of Chicago, IL) were digested with HindIII/EcoRI enzymes to isolate the genes which were then cloned into the corresponding sites in the pIZ/V5-His vector. To express LUC in invertebrate cells, the firefly luc gene from pGL3-Basic (Promega) was first amplified using primers: 5'-GGATCCATGGAAGACGCCAAAAACA-3' & 5'-TCTAGAATTACACGGCGATCTTTCCG-3', Topo-cloned, and then digested out of the Topo vector with BamHI/XbaI and cloned into BamHI/XbaI-digested p166, creating p166-LUC.

## Antibodies, immunoblotting, immunoprecipitation and multidimensional protein identification technology

Rabbit anti-Flag, mouse anti-Flag, mouse anti-tubulin, FITC-conjugated mouse anti-tubulin, and mouse anti-actin were from Sigma. Secondary antibodies used in IF microscopy were raised in donkey or goat and conjugated to Alexa 488, 568, or 647 were commercially obtained (Invitrogen). In some cases, FITC-conjugated anti-tubulin antibodies were used after secondary antibody staining with Alexa 568 and 647 antibodies for 4-color imaging. Rabbit anti-tubulin antibody was from Cell Signaling Technology (Danvers, MA). Mouse anti-LUC and rabbit anti-LUC antibodies were from Invitrogen and Abcam (Cambridge, MA), respectively. Mouse anti-VACV I3L and F4L antibodies (*Gammon et al., 2010*) were from Dr David Evans (University of Alberta, Canada). Mouse anti-VACV A27L and L1R antibodies were obtained through the NIH Biodefense and Emerging Infections Research Resources Repository (NIAID, NIH). Rabbit anti-VACV antibody was from ViroStat (Portland, ME). Mouse antibodies against VSV nucleocapsid and matrix proteins (*Lefrancois and Lyles, 1982*) were from Dr Douglas Lyles (Wake Forest School of Medicine, Winston–Salem, NC). Mouse anti-SINV E1 protein antibodies were from Dr Dianne Griffin (Johns Hopkins Bloomberg School of Public Health, Baltimore, MD).

Protein extracts for immunoblots were prepared from cell cultures by lysing cells in either reporter lysis buffer as described above or in NP-40 buffer containing 150 mM NaCl, 20 mM Tris (pH 8.0), 1 mM EDTA, and 0.5% NP-40 along with phenylmethylsulfonyl fluoride (100 µg/ml) and protease inhibitor tablets (Roche, Indianapolis, IN) as described (*Gammon et al., 2010*). Lysates were subjected to SDS-PAGE and subsequently blotted with appropriate primary antibodies after transfer to nitrocellulose membranes. Membranes were scanned using an Odyssey Infrared Imaging System (Li-COR Biosciences, Lincoln, NE).

For MuDPIT experiments, LD652 cell lysates were prepared in NP-40 buffer after 48 hr of transfection with either empty p166 vector or vector encoding Flag-A51R (control). Equal quantities of each lysate were subjected to immunoprecipitation in NP-40 buffer containing protease-inhibitor cocktail (Roche) using anti-Flag antibodies and protein G Dynabeads (Invitrogen). Immunoprecipitates were eluted from Dynabeads using glycine-HCl (pH 2) treatment for 10 min and were subsequently neutralized with neutralization buffer (0.5 M Tris–HCl, 1.5M NaCl). MuDPIT analyses of these eluted fractions were performed using an Accela HPLC and a Thermo LTQ connected to a homemade electrospray stage. Protein identification was performed with Integrated Proteomics Pipeline–IP2 (Integrated Proteomics Applications, Inc., San Diego, CA. http://www.integratedproteomics.com/). Tandem mass spectra were extracted from raw files using RawExtract 1.9.9 (*McDonald et al., 2004*), searched against the *Bombyx mori* UniprotKB proteome (as well as Flag-A51R sequence), reversed sequenced using ProLuCID, (*Peng et al., 2003*), and peptide candidates were filtered using DTASelect with the parameters -p 1 -y 1 --trypstat --sfp 0.01 -in *Tabb et al. (2002)* and *McDonald et al. (2004)*.

Radiolabeling of LD652 cell cultures (~$10^5$ cells/well) was performed 72 hr post-infection with the indicated strains by incubation of cultures with 100 µCi/well of [$^{35}$S]methionine [in methionine-free Sf900-II medium (Invitrogen)] for 2 hr. After the 2 hr pulse, [$^{35}$S]methionine-containing medium was replaced with normal growth medium and lysates were then extracted either immediately (T = 0) or at the indicated times post-pulse using radioimmunoprecipitation buffer (150 mM NaCl, 1.0% NP-40, 0.5% sodium deoxycholate, 0.1% SDS, and 50 mM Tris, pH 8.0). These whole cell extracts were then subjected to IP with anti-VSV N (10G4) antibodies or used directly in immunoblots to determine total VSV N protein levels. [$^{35}$S]-labeled VSV N IP complexes were separated by SDS-PAGE and then either dried on Whatman chromatography paper or directly transferred to nitrocellulose membranes. Radiolabeling was detected using either a Bio-Rad Personal Molecular Imager System with Quantity One software or a FLA-5000 Imaging System (Fujifilm Tokyo, Japan) equipped with Multi Gauge v3.2 software. Half-life determinations were made from non-linear regression analyses using GraphPad Prism v6 software (La Jolla, CA, USA).

## Fluorescence microscopy

Live images were captured using a Nikon Eclipse TE200 inverted fluorescent microscope with Spot Software (Diagnostics Instruments Inc., Sterling Heights, MI, version 4.6). ImageJ v1.04g software (NIH, Bethesda, MD) was used to set image thresholds and calculate the frequency of GFP-positive cells.

BSC-40 cells were split into 24-well dishes containing 12 mm glass coverslips. LD652 cells were resuspended in the growth medium and plated on 12 mm glass coverslips treated with concanavalin A (Sigma) solution (0.5 mg/ml) for 45 min prior to fixation. Cells were either fixed in 4% paraformaldehyde or methanol. Coverslips were incubated 1 hr in blocking buffer (0.5% triton X-100, 1% BSA, in PBS) prior to antibody staining. Primary and secondary antibodies were diluted in blocking buffer and incubated with rocking for 1 hr at room temperature. After antibody incubation, coverslips were washed 3–4 times with blocking buffer and mounted using ProLong Gold Antifade with DAPI (Invitrogen).

Confocal images were captured using a Nikon TE2000-E confocal microscope with MetaMorph v7.7.4 software (Molecular Devices, Sunnyvale, CA). Point spread functions were calculated using the ImageJ plugin Diffractive PSF 3D plugin (OptiNav Inc., Bellevue, WA) and deconvolved using the Iterative Deconvolve 3D plugin (OptiNav Inc). Video files were generated from confocal microscopy Z-stacks using MetaMorph v7.7.4 software.

## Animal studies

Intranasal infections were performed under anesthesia using ketamine/xylazine in saline. 5 week-old female NMRI mice (Laboratoire Elevage Janvier, Le Genest-ST-Isle, France) were inoculated with 25 µl of PBS (mock-infected group) or with 25 µl of PBS containing virus inoculum at the indicated doses (PFU). Body weight, morbidity and mortality were monitored for 20–28 days. When necessary, animals were euthanized by administering pentobarbital sodium. To determine the extent of viral replication, 5 mice per group were euthanized at day 4 and 6 post-infection, and tissues were collected and processed as previously described (*Duraffour et al., 2013*). Viral loads in tissue homogenates were titrated on HEL cells. Survival curves between VACV-WR and ΔA51R virus-infected groups were compared using log-rank (Mantel–Cox) tests and tissue virus titers were compared using unpaired Mann Whitney tests. Statistical tests were performed using GraphPad Prism v6 software.

## Acknowledgements

We thank Mr. Nathan Grindle and Dr. Paul Furcinitti for excellent technical assistance.

## Additional information

### Funding

| Funder | Grant reference number | Author |
| --- | --- | --- |
| National Institutes of Health | AI60025 | Neal Silverman |
| Howard Hughes Medical Institute | | Craig C Mello |
| National Institute of General Medical Sciences | 8 P41 GM103533 | James J Moresco, John R Yates |
| Natural Sciences and Engineering Research Council of Canada | | Don B Gammon |
| Alberta Innovates - Health Solutions | | Don B Gammon |

The funders had no role in study design, data collection and interpretation, or the decision to submit the work for publication.

### Author contributions

DBG, Conception and design, Acquisition of data, Analysis and interpretation of data, Drafting or revising the article; SD, DKR, MES, CCW, JJM, Acquisition of data, Analysis and interpretation of data; HH, RS, YC, GA, JRY, Acquisition of data, Drafting or revising the article; JHC, DC, Analysis and interpretation of data, Drafting or revising the article; DEG-R, Acquisition of data, Drafting or revising the article, Contributed unpublished essential data or reagents; WLM, NS, Analysis and interpretation

of data, Drafting or revising the article, Contributed unpublished essential data or reagents; CCM, Conception and design, Drafting or revising the article

## Ethics

Animal experimentation: All animal work was approved by the Katholieke Universiteit Leuven Ethics Committee for Animal Care and Use (Permit number: P044-2010) and all animal guidelines and policies were in accordance with the Belgian Royal Decree of 14 November 1993 and the European Directive 86-609-EEC.When necessary, animals were euthanized by administering pentobarbital sodium.

## Additional files

### Supplementary files

• Supplementary file 1. *L. dispar* transcript sequences identified by mRNA-seq and primers used for dsRNA-mediated RNAi of *L. dispar* transcripts.

• Supplementary file 2. Primers used for dsRNA-mediated RNAi of VACV transcripts.

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
