## [Decision Letter]

Thank you for sending your work entitled “A single vertebrate DNA virus protein disarms invertebrate immunity to RNA virus infection” for consideration at *eLife*. Your article has been favorably evaluated by Detlef Weigel (Senior editor) and 2 reviewers, one of whom is a member of our Board of Reviewing Editors, and one of whom, Rollie Clem, has agreed to reveal his identity.

The Reviewing editor has assembled the following comments to help you prepare a revised submission.

The authors of this study make a surprising discovery that VACV infection of a Lepidopteran cell line can rescue infection with two RNA viruses (VSV and SINV). The authors identify a previously uncharacterized VACV gene product A51R as being responsible for the rescue of RNA viral infection. The mechanism of A51R effect is complex, as it associates with such polyfunctional targets as microtubules and ubiquitin. A51R is shown to promote viral protein stability, possibly by interfering with Ub-mediated degradation. VACV lacking functional A51R is attenuated and this may be an important feature of the vaccine starin of the virus, which has a truncation of A51R.

The manuscript has several strengths: it develops a new experimental system that will benefit many investigators: the LD652 cell line will likely become a system of choice to investigate antiviral immunity in Lepidoptera. The authors also identify multiple targets for A51R, which will be useful for future detailed characterizations of its mechanisms of action, which likely to be very complex.

Overall the manuscript reports novel insights into the nature of RNA virus restriction in lepidopteran cells. The findings are an important contribution to our knowledge of anti-viral responses in these organisms. The mechanism(s) by which A51R relieves the restriction of VSV and SINV is still vague, but there are important clues that can be followed in future work.

From a broader biological perspective, the findings suggest that poxvirus infections may generally enable co-infections with RNA viruses, perhaps in species outside of Lepidoptera order of insects. Because VACV is unlikely to benefit from a concomitant RNA virus infection, one possibility is that A51R evolved to perform function(s) that have nothing to do with RNA viruses, but rather RNA viruses evolved to exploit the altered host cells (due to VACV infection) to establish a productive infection. From this perspective, it would be very informative to examine the consequence of VACV–VSV co-infection in mice, a prediction being that this co-infection will have a strong synergy (increased virulence or viral load).

Major points of concern:

1) The authors report that VSV and SINV do not replicate in Ld652 cells, however co-infection with vaccinia allowed VSV and SINV to replicate. The block appears to be at a post-entry step, because immunostaining revealed the presence of viral particles in punctate staining pattern inside cells at 24 hpi. I believe the authors have missed an important clue: in the case of SINV, their antibody was against the E1 protein, an envelope protein of the virus. The observation of E1 staining in a punctate pattern inside the cell at 24 hpi indicates that the virus has been taken up by endocytosis but is trapped in endosomes. This is the only way to explain why the E1 protein, a transmembrane protein, is present inside the cells. Thus, A51R may act at least in part by promoting release of these viruses from endosomes. This could potentially involve microtubule trafficking and/or ubiquitin-mediated proteolysis, and could perhaps bring increased clarity to their overall findings. The authors need to re-examine their findings in light of this observation.

2) The authors conclude that A51R may act in part by promoting the stability of viral proteins, but the evidence here is not convincing. After a 2 hr pulse with labeled methionine, the levels of labeled VSV N protein were similar in cells infected with either wt vaccinia or a mutant lacking A51R, but immunoblotting indicated that the overall levels of N were substantially lower in the mutant-infected cells. However this could be due to the 100-fold decreased replication of VSV in cells infected with A51R mutant vs wt vaccinia (Figure 4). In order to conclude that A51R affects the stability of N, it would be necessary to do a proper pulse-chase experiment to properly determine the half life of N in cells infected with wt vs A51R mutant vaccinia (and preferably also in cells transfected with A51R in the absence of other viral genes).

---

## [Author Response]

*1) The authors report that VSV and SINV do not replicate in Ld652 cells, however co-infection with vaccinia allowed VSV and SINV to replicate. The block appears to be at a post-entry step, because immunostaining revealed the presence of viral particles in punctate staining pattern inside cells at 24 hpi. I believe the authors have missed an important clue: in the case of SINV, their antibody was against the E1 protein, an envelope protein of the virus. The observation of E1 staining in a punctate pattern inside the cell at 24 hpi indicates that the virus has been taken up by endocytosis but is trapped in endosomes. This is the only way to explain why the E1 protein, a transmembrane protein, is present inside the cells. Thus, A51R may act at least in part by promoting release of these viruses from endosomes. This could potentially involve microtubule trafficking and/or ubiquitin-mediated proteolysis, and could perhaps bring increased clarity to their overall findings. The authors need to re-examine their findings in light of this observation*.

We thank the reviewers for this insightful comment. It is indeed possible that part of the restriction to both VSV and SINV may be due to a poor ability of these viruses to escape from endosomes upon entry. While this is an excellent point that may highlight another layer of Lepidopteran cell restriction to infection, we feel that clearly defining a role for endosome trapping in RNA virus restriction in this new system is beyond the scope of this initial report. Furthermore, it is clear that at least some virus particles escape the endosome in the absence of A51R because 1) some RNA transcription and LUC reporter gene expression can be detected during single VSV-LUC/SINV-LUC infections (Figure 1—figure supplement 1), and 2) knockdown of *L. dispar* RNAi factors such as AGO2/Dicer-2 (Figure 2), which would target intracellular viral transcripts during replication, partially relieve the restriction of VSV/SINV in LD652 cells.

Because we believe that endosome trapping may be important for future studies, we did attempt initial experiments to investigate co-localization of VSV/SINV markers with endosomes. Given the lack of validated antibodies to *L. dispar* endosome proteins, we screened several polyclonal antibodies raised against endosome marker proteins (e.g., rabenosyn-5, early endosome antigen 1 etc) for cross reactivity with homologs in LD652 cells. Unfortunately we were unable to observe a clear, punctate staining pattern with any of these antibodies indicative of endosome staining. We also tried labeling all membranous vesicles in LD652 cells with a fluorescent lipohilic dye (FM^™^ 4-64FX). However, in our hands, we found that this fluorescent dye was lost during our fixation/immunostaining procedure for viral antigens, despite trying various fixatives and permeabilization techniques. Thus, the inability of our antibodies and FM^™^ 4-64FX dye to mark *L. dispar* endosomes has made it clear to us that we will be unable to address the endosome question until we find suitable reagents for *L. dispar*.

We have included a paragraph in the Discussion to discuss the possibility that VSV/SINV virions become trapped in endosomes in LD652 cells. We have also discussed the possibility that A51R might be involved in the known perturbation of endosomal trafficking during VACV infection (62) and that future studies should investigate if A51R-mediated perturbation of MT-dependent endosomal trafficking may affect VSV/SINV replication in *L. dispar* cells.

*2) The authors conclude that A51R may act in part by promoting the stability of viral proteins, but the evidence here is not convincing. After a 2 hr pulse with labeled methionine, the levels of labeled VSV N protein were similar in cells infected with either wt vaccinia or a mutant lacking A51R, but immunoblotting indicated that the overall levels of N were substantially lower in the mutant-infected cells. However this could be due to the 100-fold decreased replication of VSV in cells infected with A51R mutant versus wt vaccinia (*Figure 4*). In order to conclude that A51R affects the stability of N, it would be necessary to do a proper pulse-chase experiment to properly determine the half life of N in cells infected with wt versus A51R mutant vaccinia (and preferably also in cells transfected with A51R in the absence of other viral genes)*.

We have included data from pulse-chase experiments (Figure 9) showing that VSV N protein stability is reduced during co-infection with the ΔA51R strain compared to VACV-WR. We have also indicated in the text that regression analyses of these pulse-chase experiments suggest that there is indeed a reduced half-life of VSV-N during co-infection with the ΔA51R strain (∼1.8 hr) when compared to co-infection with VACV-WR (∼5.3 hr). We made further attempts to conduct pulse-chase experiments of VSV-N in LD652 cells transfected with either Flag-GFP (as a control) or Flag-A51R vectors but found (as in Figure 9) that radiolabeled VSV N signals were too low in the control treatments to calculate half-lives to compare to the Flag-A51R treatments. Thus, using this approach we were unable to distinguish defects in VSV protein translation rates vs protein stability. Therefore, we opted to only include the pulse-chase data obtained from co-infection experiments. The “residual” VSV rescue afforded by the ΔA51R co-infection (presumably due to other factors encoded by VACV) allows one to actually detect radiolabeled VSV N protein in the absence of A51R and thus one can directly compare half-lives in these conditions to the VACV-WR co-infections.